# Optogenetically induced low-frequency correlations impair perception

Anirvan Nandy[1,2]*, Jonathan J Nassi[1†], Monika P Jadi[2,3], John Reynolds[1]

[1]Systems Neurobiology Laboratories, The Salk Institute for Biological Studies, La Jolla, United States; [2]Department of Neuroscience, Yale University, New Haven, United States; [3]Department of Psychiatry, Yale University, New Haven, United States

**Abstract** Deployment of covert attention to a spatial location can cause large decreases in low-frequency correlated variability among neurons in macaque area V4 whose receptive-fields lie at the attended location. It has been estimated that this reduction accounts for a substantial fraction of the attention-mediated improvement in sensory processing. These estimates depend on assumptions about how population signals are decoded and the conclusion that correlated variability impairs perception, is purely hypothetical. Here we test this proposal directly by optogenetically inducing low-frequency fluctuations, to see if this interferes with performance in an attention-demanding task. We find that low-frequency optical stimulation of neurons in V4 elevates correlations among pairs of neurons and impairs the animal's ability to make fine sensory discriminations. Stimulation at higher frequencies does not impair performance, despite comparable modulation of neuronal responses. These results support the hypothesis that attention-dependent reductions in correlated variability contribute to improved perception of attended stimuli.
DOI: https://doi.org/10.7554/eLife.35123.001

*For correspondence: anirvan.nandy@yale.edu

Present address: †Inscopix Inc, Palo Alto, United States

Competing interests: The authors declare that no competing interests exist.

## Introduction

Neurons exhibit responses that are highly variable (*Shadlen and Newsome, 1998*), with nearby neurons in the cortex exhibiting correlated variability in their spiking output (*Cohen and Kohn, 2011*; *Smith and Kohn, 2008*; *Smith and Sommer, 2013*; *Zohary et al., 1994*). It has been estimated on theoretical grounds that even weak correlations substantially reduce the information coding capacity of a population (*Zohary et al., 1994*). Spatial attention can reduce correlated variability (often referred to as noise correlations) among neurons in macaque visual area V4 (*Cohen and Maunsell, 2009*; *Mitchell et al., 2009*), an area that is strongly modulated by spatial attention (*Reynolds and Chelazzi, 2004*). *Mitchell et al., 2009* found that this reduction is restricted to low frequencies below 10 Hz. These studies have estimated, on theoretical grounds, that the reduction in correlated variability accounts for a large fraction (about 80%) of the perceptual benefit due to attention. However, these estimates rely on specific assumptions about the relationship between noise and signal correlations, and thereby, on how population signals are read out in the brain (*Abbott and Dayan, 1999*; *Averbeck et al., 2006*; *Moreno-Bote et al., 2014*; *Panzeri et al., 1999*). Theoretical studies using heterogeneous tuning curves and optimal readout have concluded that correlated variability does not necessarily limit information (*Ecker et al., 2011*; *Shamir and Sompolinsky, 2006*). Consistent with the interpretation that attention-dependent reductions in correlated variability improve perception, a recent study of the effects of naturally occurring fluctuations in neural correlations found improved sensory discrimination when neurons in Area V4 were desynchronized (*Beaman et al., 2017*). However, other studies have posited that correlations themselves may be induced by fluctuations in attention (*Goris et al., 2014*), resulting in variation in response gain that is

shared across neurons, and other experiments have shown that attention can, under some conditions, also increase correlated neural response variability (*Ruff and Cohen, 2014*). Taken together, these studies call into question the simple idea that attention reduces correlations so as to improve sensory discrimination. Importantly, all of these studies are correlative in nature. The causal role of correlated variability in perception has not been tested and thus the proposal that low-frequency correlated variability is detrimental for perception has remained purely hypothetical.

Here, we sought to directly test the effects of correlated variability on sensory discrimination by using optogenetic activation to induce correlations in Area V4 as monkeys performed an orientation discrimination task near perceptual threshold. We exploited the fact that attentional modulation of correlated variability is both spatially- and frequency selective: attention-dependent reductions in correlation are restricted to low frequencies (<10 Hz (*Mitchell et al., 2009*)). We reasoned that the correlations that impair perception may have an inherent time scale, with low- but not high-frequency correlations impairing perception. If so, we would predict that the effects of correlations on perception should be specific to this low-frequency range.

## Results and discussion

We took advantage of a novel approach to primate optogenetics and electrophysiology (*Nassi et al., 2015*; *Ruiz et al., 2013*) in which the native dura mater is replaced by a silicone based artificial dura (*Figure 1A, B*). This approach provides an optically clear window into the awake-behaving primate brain and allows precise opto-electrophysiology. We injected a lenti-viral construct (lenti-CaMKIIa-C1V1$_{E162T}$-ts-EYFP) to preferentially drive expression of the depolarizing opsin C1V1 in excitatory neurons (*Yizhar et al., 2011*) in a restricted portion (200-300 μm diameter) of dorsal V4 of two macaque monkeys (*Figure 1C*). Despite some heterogeneity in orientation tuning width at each injection site, overall there was similar tuning among neurons within a site (*Figure 2—figure supplement 1*).

We trained two monkeys to perform an attention-demanding orientation-change detection task (*Figure 2A*). The monkeys were spatially cued to attend to one of two spatial locations. In the 'attend in' condition, the monkeys were instructed to covertly attend to a spatial location within the

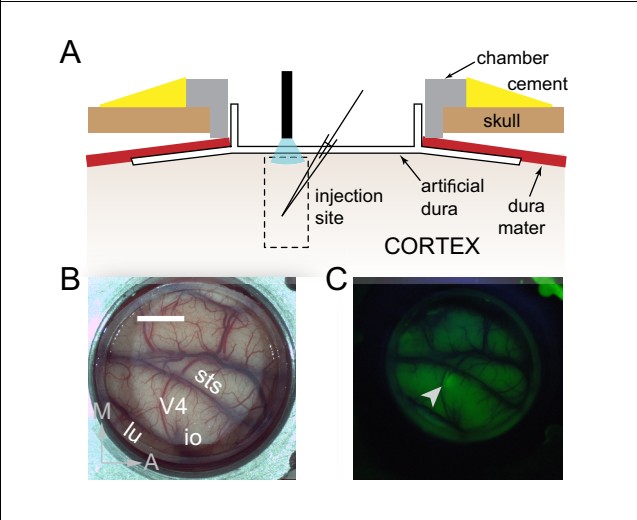

**Figure 1.** Surface Optogenetics and electrophysiology through an artificial dura. (**A**) Schematic of an artificial dura (AD) chamber. A portion of the native dura mater (red) is resected and replaced with a silicone based optically clear artificial dura (AD). The optical clarity of the AD allows precisely targeted injections of viral constructs and subsequent optical stimulation and electrophysiological recordings. (**B**) An AD chamber is shown over dorsal V4 in the right hemisphere of Monkey A. sts = superior temporal sulcus, lu = lunate sulcus, io = inferior occipital sulcus. Area V4 lies on the pre-lunate gyrus between the superior temporal and lunate sulci. Scale bar = 5 mm; M = medial, A = anterior (**C**) EYFP expression at the first injection site (lenti-CaMKIIα-C1V1-ts-EYFP) after 4 weeks. DOI: https://doi.org/10.7554/eLife.35123.002

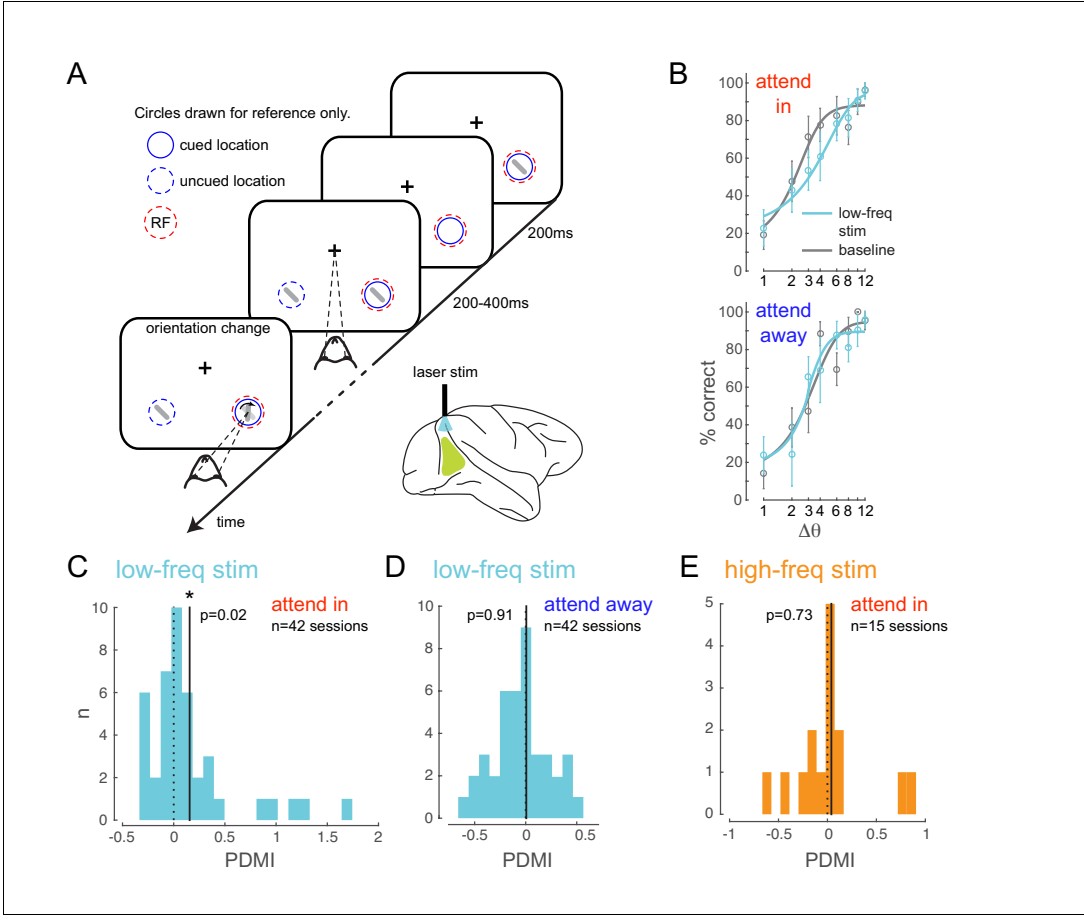

**Figure 2.** Optogenetically-induced low-frequency correlations cause a frequency- and spatially-selective impairment in an attention-demanding orientation discrimination task. (**A**) Attention task: While the monkey maintained fixation, two oriented Gabor stimuli (schematized as oriented bars) flashed on and off simultaneously at two spatial locations: one at the RF of the opsin injection site, the other at a location of equal eccentricity across the vertical meridian. The monkey was cued to covertly attend to one of the two locations. At an unpredictable time, one of the two stimuli changed in orientation. The monkey was rewarded for making a saccade to the location of orientation change at either location (95% probability of change at cued location; 5% probability at un-cued location [foil trials]). If no change occurred (catch trials), the monkey was rewarded for maintaining fixation. On a random subset of trials, the opsin site was optically stimulated using a low-frequency (4-5 Hz) sinusoidally modulated laser light ($\lambda = 532$nm). (**B**) Psychometric functions for an example behavioral session showing performance (hit rate) as a function of task difficulty (size of orientation change) for the baseline (no optical stimulation) condition in gray and low-frequency optical stimulation condition in blue. *Top*, monkey was instructed to attend to the site of optical stimulation; *Bottom*, monkey was instructed to attend to the contralateral hemifield. Error bars are std. dev. obtained by a jackknife procedure and corrected for the number of jackknives (20). The data has been fitted with a smooth logistic function. (**C**) The perceptual discrimination modulation index (PDMI; change in psychometric function threshold due to optical stimulation) in the low-frequency optical stimulation condition when the monkey was attending in to the site of optical stimulation across all behavioral sessions. The solid line represents the mean of the distribution. The PDMI distribution is significantly different from zero. (**D–E**) No significant change in PDMI either when the monkey was attending away from the site of optical stimulation (**D**) or due to high-frequency optical stimulation (**E**).

DOI: https://doi.org/10.7554/eLife.35123.003

The following figure supplements are available for figure 2:

**Figure supplement 1.** Orientation tuning properties at opsin injection sites.

DOI: https://doi.org/10.7554/eLife.35123.004

**Figure supplement 2.** Behavioral performance.

DOI: https://doi.org/10.7554/eLife.35123.005

**Figure supplement 3.** Behavioral changes with optical stimulation.

*Figure 2 continued*

DOI: https://doi.org/10.7554/eLife.35123.008
**Figure supplement 4.** Other control conditions.
DOI: https://doi.org/10.7554/eLife.35123.006
**Figure supplement 5.** Irradiance response curves.
DOI: https://doi.org/10.7554/eLife.35123.007

receptive fields of neurons at the viral injection site, while maintaining fixation at a central fixation point. In the 'attend-away' condition attention was directed to a location of equal eccentricity across the vertical meridian. On each trial, a sequence of oriented Gabor stimuli simultaneously flashed on and off at both spatial locations (200 ms on, variable 200–400 ms off). At an unpredictable time (minimum 1 s, maximum 5 s), one of the two stimuli (95% probability at cued location; 5% probability at uncued location, 'foil trials') briefly changed in orientation (200 ms) and the monkey was rewarded for making a saccade to the location of orientation change. If no change occurred within 5 s ('catch trials', 13% of trials), the monkey was rewarded for holding fixation. We controlled task difficulty by varying the degree of orientation change and thereby obtained behavioral performance curves (psychometric functions) for each recording session (*Figure 2B*, *Figure 2—figure supplement 2A,B*). Impaired performance (*Figure 2—figure supplement 2A*, left panel, square symbol) and slower reaction times (*Figure 2—figure supplement 2A*, right panel, square symbol) were observed for the foil trials, indicating that the monkey was indeed using the spatial cue in performing the task.

To test if low-frequency correlations impair discrimination we optically stimulated neurons at the opsin injection site with 4-5Hz sinusoidally modulated low-power laser stimulation, on a randomly chosen subset of trials ('low-frequency stimulation' condition). Our goal was to induce correlations without significantly altering the mean firing rates by using low-power stimulation. Significant changes in mean firing rate could have unknown effects such as masking of the stimulus evoked response. Equating firing rates also avoids any indirect effects of mean firing rate changes on spike-count correlations (*Cohen and Kohn, 2011*). We find that low-frequency optical stimulation modulates the timing of the neural response (Figure 4D) but does not alter the overall magnitude of the population response (Figure 4A, B, C). We replicate previous findings that attention reduces low-frequency spike-count correlations in the baseline (no optical stimulation) condition (*Mitchell et al., 2009*) (*Figure 3A*, left panel; gray versus white bars, $p = 0.02$, $t$-test). As predicted, low-frequency optical stimulation increases low-frequency correlations (*Figure 3A*, left panel; blue versus gray bar, $p = 0.045$, $t$-test). The induced correlations were at a level comparable in strength to that observed when attention was directed away from the RF location in the baseline condition (*Figure 3A*, left panel; blue versus white bar). Optogenetic activation is accompanied by a period of reduced activity following stimulation. By careful titration of laser intensity (amplitude of sinusoidal modulation) we were able to alter the timing of spiking without altering mean firing rate. This is shown in *Figure 4*: we see a robust increase in firing rate due to attention in both the low-frequency stimulation and baseline conditions (*Figure 4A, B*), but there is no significant rate increase due to optical stimulation either during the pre-stimulus blank period (*Figure 4C*, left panel, $p = 0.49$, $t$-test; *Figure 4—figure supplement 1*, top-left panel) or during the stimulus presentation period (*Figure 4C*, right panel, $p>0.1$, $t$-test; *Figure 4—figure supplement 1*, bottom-left panel). Rather, unit activity shows phase locking to optical stimulation (*Figure 4D,E*; pre-stimulus period). The distributions of spiking activity with respect to the phase of the optical stimulation show significant deviation from what would be expected from a null distribution (*Figure 4D*, example units, $p \ll 0.01$; *Figure 4E*, population, $p<0.01$, Rayleigh test). The null distributions were derived from a rate-matched Poisson process. We see a similar phase locking due to optical stimulation during the stimulus presentation period (*Figure 4F*, $p<0.01$, Rayleigh test), although the peak of the phase-lock distribution for the stimulus presentation period occurs earlier (around 120°) compared to that for the pre-stimulus period (around 210°). This would be expected, as the neurons are depolarized by the visual stimulus and hence more easily pushed to spiking threshold by optogenetic depolarization, as compared to when no stimulus is present. Thus, the physiology data shows that we successfully induced correlated activity among neurons at the opsin site without affecting the response rates.

Behaviorally, we find that low-frequency stimulation impairs the monkey's ability to detect fine orientation changes, and does so only at the opsin location, as indicated by impairment in the

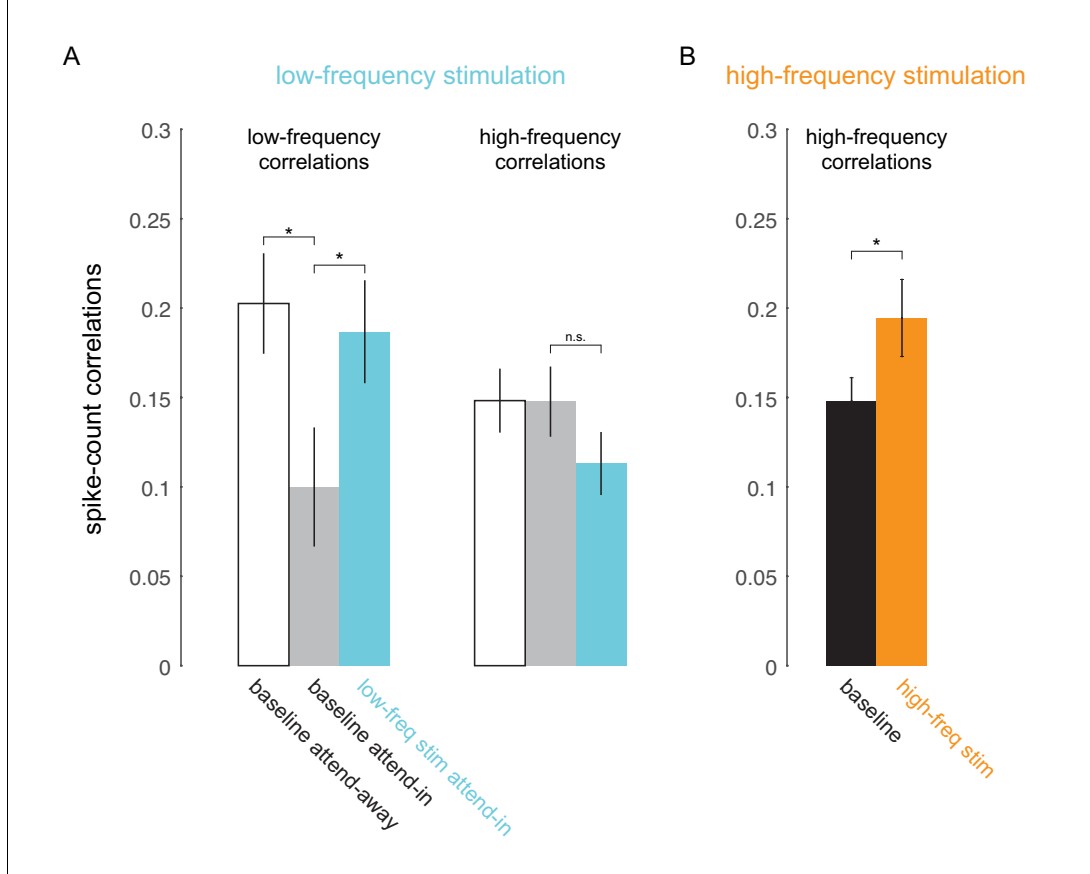

**Figure 3.** Optical stimulation at low- and high-frequencies induces low- and high-frequency correlated activity. (**A**) Consistent with earlier reports (*Mitchell et al., 2009*), attention reduces baseline spike-count correlations at low frequencies (200ms counting window, $p = 0.02$; left panel, white versus gray bar) but not at high frequencies (50ms window; right panel, white versus gray bar). Low-frequency optical stimulation increases low-frequency correlations ($p = 0.045$; left panel, gray versus blue bar) but not high-frequency correlations ($p > 0.1$; right panel, gray versus blue bar). (**B**) High-frequency optical stimulation increases high-frequency correlations ($p < 0.05$). $n = 79$ pairs for baseline and low-frequency stimulation, $n = 27$ pairs for high-frequency stimulation, collapsed across attention conditions. Mean +/- s.e.m. in all plots.
DOI: https://doi.org/10.7554/eLife.35123.009

attend-in condition (*Figure 2B*, upper panel; *Figure 2—figure supplement 2B*), not in the attend-away condition (*Figure 2B*, lower panel), in which the monkey discriminated orientation at the contralateral location. To quantify this behavioral deficit, we estimated the threshold of the monkeys' psychometric functions and calculated the change in threshold due to optical stimulation as a modulation index (perceptual discrimination modulation index, PDMI; see Materials and methods). We find a significant increase in PDMI due to low-frequency stimulation in the attend-in condition ($p = 0.02$, *t*-test; *Figure 2C*), indicating impaired detection of fine orientation changes. Unexpectedly, we also found a significant increase in slope ($p = 0.007$, *t*-test; *Figure 2—figure supplement 5A*), suggesting that the shift from non-detection to detection occurs over a narrower range of orientations in the laser stimulation condition. In a large fraction of individual behavioral sessions (*Figure 2—figure supplement 5B*), both the changes in threshold and slope were significant (20/42 sessions). 11/42 sessions had significant threshold change only, while 5/42 sessions had significant slope change only.

The impairment due to optical stimulation is location specific: there was no significant change in PDMI on trials when the monkey was cued to detect the target at the unstimulated location (attend-away condition, *Figure 2D*). Importantly, the impairment is also frequency specific. When we stimulate the neurons with 20Hz sinusoidally modulated low-power laser stimulation ('high-frequency stimulation' condition), we observed no significant change in PDMI (*Figure 2E*), despite a significant increase in high-frequency spike-count correlations (*Figure 3B*) and phase locking comparable to the

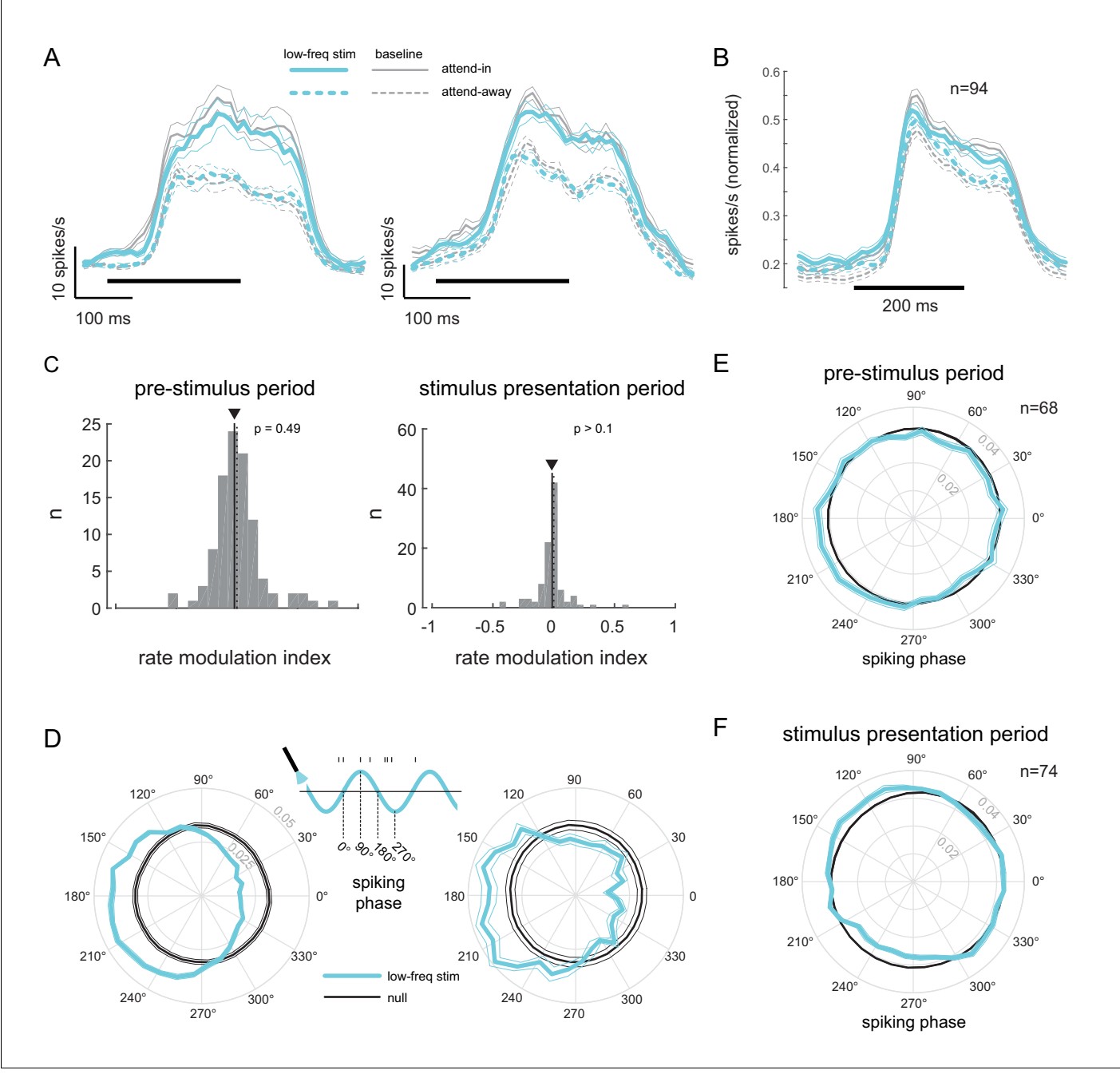

**Figure 4.** Low-frequency stimulation induces phase-locking without increasing firing rates. (A) Peri-stimulus time histograms (PSTH) of two example units for the different experimental conditions. Both units show a robust firing rate modulation due to attention (solid versus dashed lines) but no rate increase due to low-frequency optical stimulation (blue versus gray lines). Horizontal bars represent stimulus duration. (B) Population data showing the same rate increase due to attention, but no significant increase due to optical stimulation ($n = 94$). Same convention as in (A). (C) Distribution of rate modulation indices for the low-frequency stimulation attend-in condition compared to the baseline attend-in condition for a 200ms pre-stimulus period (left panel) and 200ms stimulus presentation period (60-260ms after stimulus onset; right panel). The arrowheads depict the median of the distributions. Neither distribution is significantly different from zero ($p>0.1$). (D) Phase plots for two example units showing the distribution of spiking activity with respect to the phase of the optical stimulation, during the pre-stimulus period. In gray is the null distribution obtained from a rate-matched Poisson process. Both units show significant deviations from the null distribution ($p \ll 0.01$ for both, Rayleigh test), indicative of phase locking. (E) Population phase-locking plot illustrating the bias in spiking activity to the downswing of optical stimulation during the pre-stimulus period ($n = 68$). Same convention as in D). The distribution of spiking phase is significantly different from null ($p<0.01$, Rayleigh test). (F) Same as in (E), but for the stimulus presentation period ($n = 74$). The distribution of spiking phase is significantly different from null ($p<0.01$, Rayleigh test).

DOI: https://doi.org/10.7554/eLife.35123.010

*Figure 4 continued on next page*

*Figure 4 continued*

The following figure supplements are available for figure 4:

**Figure supplement 1.** Comparison of spike rates between baseline and optical stimulation.

DOI: https://doi.org/10.7554/eLife.35123.011

**Figure supplement 2.** Phase-locking to optical stimulation.

DOI: https://doi.org/10.7554/eLife.35123.012

**Figure supplement 3.** Behavioral performance is not affected by optical stimulation phase.

DOI: https://doi.org/10.7554/eLife.35123.013

**Figure supplement 4.** Optical stimulation does not change orientation tuning.

DOI: https://doi.org/10.7554/eLife.35123.014

**Figure supplement 5.** Optical stimulation does not cause frequency-specific adaptation.

DOI: https://doi.org/10.7554/eLife.35123.015

low-frequency stimulation condition (*Figure 4—figure supplement 2*). As in the low-frequency stimulation condition, we find no significant changes in mean firing rates with the high-frequency stimulation condition (*Figure 4—figure supplement 1*, right panels). The stimuli in each sequence were presented with irregular timing to ensure that any impairment did not stem from stimuli appearing at a particular phase of the laser stimulation, such as the phase at which neural sensitivity was at its nadir. To verify that such phase alignment did not nonetheless occur by chance, we measured the phase of target stimulus onset for both low- and high-frequency stimulation and found no phase preference (*Figure 4—figure supplement 3A*). Nor was behavioral performance phase dependent (*Figure 4—figure supplement 3B*). A two-way ANOVA of normalized performance with factors 'laser-phase' (two different phase bin arrangements are shown in *Figure 4—figure supplement 3B*) and 'delta-orientation' (the trial-by-trial difference between target and non-target orientation) revealed no significant main effect of laser phase ($F_{\text{laser−phase}} = 1.53$, $p = 0.2$, left bin arrangement in *Figure 4—figure supplement 3B*; $F_{\text{laser−phase}} = 0.73$, $p = 0.53$, right bin arrangement in *Figure 4—figure supplement 3B*), a significant main effect of orientation and no significant interaction between the factors. A second potential concern is that the laser might impair orientation discrimination by distorting or flattening orientation tuning curves. We find that orientation tuning curves were not significantly altered by the laser at either frequency (*Figure 4—figure supplement 4*), at least in the range of orientations used in the experiment. A two-way ANOVA of normalized firing rates with factors 'laser-condition' (low-frequency laser, no laser) and 'orientation' revealed no significant main effect of laser condition ($F_{\text{laser−condition}} = 0.8$, $p = 0.37$), a significant main effect of orientation and no significant interaction between the two factors. Similarly, we found no significant main effect of the high-frequency laser condition ($F_{\text{laser−condition}} = 0.05$, $p = 0.82$), a significant main effect of orientation and no significant interaction between the two factors. A third potential concern is that the rhythmic laser stimulation might cause a sort of frequency-dependent adaptation that would cause neurons to be less sensitive to visual stimuli presented at a similar frequency. If so, the low-frequency (4-5Hz) laser stimulation could reduce the responses evoked by 200ms visual stimuli, impairing the monkey's ability to discriminate the stimuli, while the high-frequency laser stimulation might not cause this effect, explaining the observed impairment. To test this, we measured the firing rates evoked by the first four non-target stimulus flashes, on no-laser, low-frequency laser and high-frequency laser trials, in the 'attend in' condition (*Figure 4—figure supplement 5*). Though the first stimulus in the sequence evoked a stronger response than the subsequent stimuli (reflecting a form of visual stimulus-driven adaptation), we find no evidence that the addition of the laser at either frequency caused a change in mean firing rate. A two-way ANOVA of normalized firing rate with factors 'flash-position' (1,2,3 or 4) and 'laser condition' (low-frequency laser, high-frequency laser, no laser) revealed a significant main effect of flash position ($F_{\text{flash−position}} = 3.33$, $p = 0.02$), no significant main effect of laser condition ($F_{\text{laser−condition}} = 0.65$, $p = 0.52$) and no significant interaction between the two factors. For the small number of sessions ($n = 15$) over which we could do this analysis, the PDMI trends toward significance for the low-frequency condition ($p = 0.07$), but is highly non-significant for the high-frequency condition ($p = 0.7$). Additionally, we did not find significant changes in false-alarm rates with either low- or high-frequency stimulation ($p>0.1$, *t*-test; *Figure 2—figure supplement 2E*). This was true for false alarms made during catch trials as well as on non-catch trials. Thus, we find no evidence that laser stimulation caused our subjects to mis-perceive a non-target as a target.

V4 has patchy organization for orientation tuning, so simultaneously recorded neurons tended to prefer similar orientations (see *Figure 2—figure supplement 1*). Under these conditions, where signal correlations are positive, positive noise correlations should reduce discriminability (*Averbeck et al., 2006*) by increasing the overlap between neural responses evoked by discriminanda. In the present experiment, where the task was to discriminate target from non-target, this would predict that in sessions where we observed laser-induced perceptual impairment, we should observe laser-induced decreases in discriminability at the level of pairs or populations of neurons, especially on miss trials, when the monkey was unable to discriminate target from non-target. For each recording session, we calculated a neural measure of discriminability between non-target and target stimuli across simultaneously recorded neurons (neural discriminability modulation index, NDMI; *Figure 5*; see Materials and methods, (*Cohen and Maunsell, 2010*)). We then examined whether laser-induced increases in perceptual threshold were correlated with laser-induced reductions in neural discriminability on trials in which the target appeared at the opsin location (the 'attend in' condition). *Figure 5B* shows NDMI for each experimental session (calculated from miss trials where the animal failed to detect a target) against the corresponding PDMI. We examined this in two ways: by measuring the correlation in a N-dimensional space where N is the number of neurons recorded in a given session (left panel) or by measuring the average NDMI across neural pairs (right panel). As predicted, there is a strong and significant negative correlation in both analyses (*Figure 5B*; $\rho = -0.29$ $(p = 0.05)$, NDMI from all simultaneous neurons, left panel; $\rho = -0.42$ $(p = 0.01)$, average NDMI across all simultaneous pairs, right panel; robust correlation). In both analyses, the sessions in which low-frequency laser-induced correlations caused the strongest perceptual suppression all showed negative NDMIs. In other words, laser-induced reductions in discriminability at the neural level corresponded to increased perceptual thresholds. There was no significant correlation between NDMI and changes in the slope of the psychometric function. NDMI calculated from hit trials had no correlation with threshold or slope changes.

To confirm whether it is possible to induce coherent activity in a neuronal ensemble due to subthreshold rhythmic stimulation, we examined the consequences of such stimulation on a conductance-based model of excitatory and inhibitory neurons (*Figure 6A*; see Materials and methods). We calculated the strength of coherent activity in the network (spike-spike coherence, SSC) both with

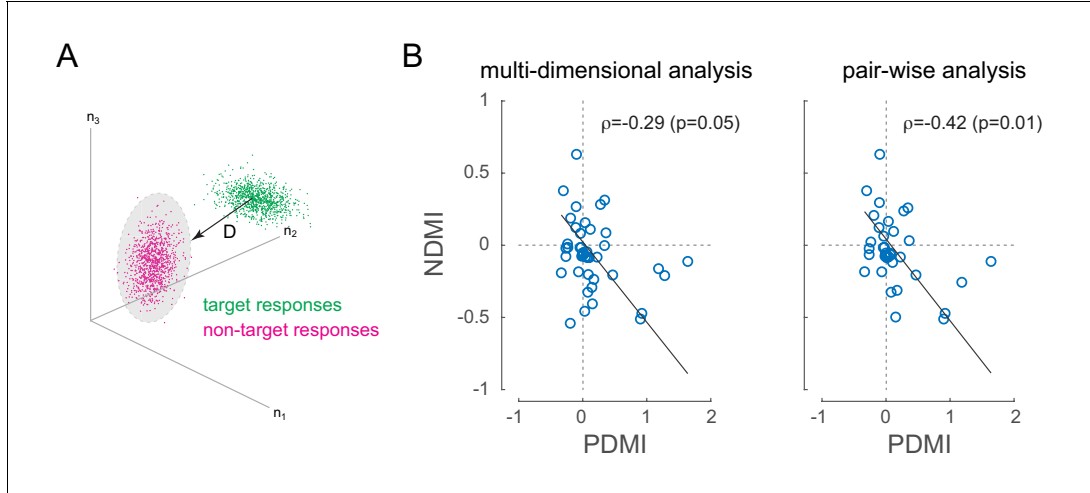

**Figure 5.** Optical modulation of neural discriminability correlates with behavioral perturbations. (**A**) Schematic of neural discriminability analysis. The responses of a hypothetical set of 3 neurons to target (green) and non-target (magenta) stimuli are depicted as point clouds. Each dot represents a stimulus presentation. The discriminability (D) between the two response categories is defined as the Mahalanobis distance between the centroid of the target responses and the non-target point cloud. (**B**) Neural Discriminability Modulation Index (NDMI) due to optical stimulation is plotted against the corresponding PDMI (behavioral threshold change) for each experimental session. NDMI is calculated either from multi-dimensional clouds from all simultaneously recorded neurons (left panel; n = 42 sessions) or as the average of two-dimensional clouds from all pairs of simultaneously recorded neurons (right panel; n = 35 sessions). NDMI is negatively correlated with PDMI. Since both NDMI and PDMI are dependent measures, the data were fitted with a line whose slope was obtained from a Model II regression.
DOI: https://doi.org/10.7554/eLife.35123.016

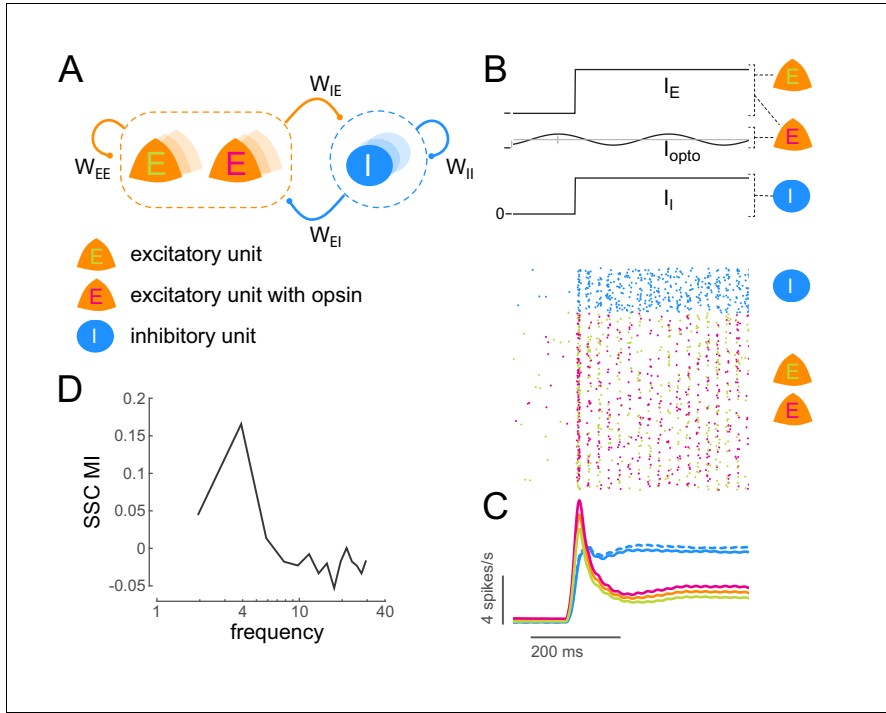

**Figure 6.** Low-frequency sub-threshold stimulation induces coherent activity in a computational model of E-I neurons. (**A**) Schematic of a local conductance-based E-I network with mutually coupled excitatory (**E**) and inhibitory (I) units. A fraction (50%) of the E units are sensitive to 'optical' stimulation. $W_{ee}$, self-excitation among E units; $W_{ii}$, self-inhibition among I units; $W_{ie}$, excitation provided by E to I; $W_{ei}$, inhibition provided by I to E. (**B**) Simulation of a network of 800 E and 200 I units ($W_{ee} = 16$, $W_{ii} = -1$, $W_{ie} = 4$, $W_{ei} = -18$). The raster plot shows the activity of all units in the model (blue, I; green, E without opsin; magenta, E with opsin) to a step input ($I_e, I_i$) and 4Hz sinusoidal optical stimulation ($I_{opto}$). (**C**) Population spiking rate averaged across 1000 simulations of the scenario in (**B**) with and without optical stimulation. (blue, I; orange, all E; green, E without opsin; magenta, E with opsin. solid lines, with optical stimulation; dashed lines, without optical stimulation) (**D**) Spike- spike coherence (SSC) among E units was calculated for the two conditions with and without optical stimulation and the change in SSC across the two conditions was calculated as a modulation index (SSC MI). SSC MI exhibits a peak at 4Hz due to optical stimulation.

DOI: https://doi.org/10.7554/eLife.35123.017

The following figure supplements are available for figure 6:

**Figure supplement 1.** Induction of coherent activity in the E-I model is robust across network and stimulation parameters - I.
DOI: https://doi.org/10.7554/eLife.35123.018
**Figure supplement 2.** Induction of coherent activity in the E-I model is robust across network and stimulation parameters - II.
DOI: https://doi.org/10.7554/eLife.35123.019

and without sub-threshold stimulation (*Figure 6B,C*). We quantified the change in coherence due to stimulation as a modulation index (SSC MI; *Figure 6D*). We find that it is indeed possible to induce coherent activity in the network at a desired frequency (*Figure 6D*, *Figure 6—figure supplement 1D*) and that this induction is robust to a wide range of network (*Figure 6—figure supplement 1C*, *Figure 6—figure supplement 2A*) and stimulation parameters (*Figure 6—figure supplement 2B*).

The location specificity of the impairment also suggests that the impairment is not due to a phosphene effect (*Jazayeri et al., 2012*). If attention were drawn away from the unstimulated location by a phosphene we would expect impaired performance in the attend-away condition. We also verified that the impairment was not due to a thermal effect by stimulating a location in the chamber a few millimeters from the opsin site (*Figure 2—figure supplement 4A*) and not due to visual distractions caused by the laser light by stimulating outside the brain (*Figure 2—figure supplement 4B*). In both cases, we did not observe any changes in behavior.

One of the two animals used in this study was euthanized to verify that, as we have previously found in macaque V1 (*Nassi et al., 2015*), lenti/CAMKII leads to selective expression in pyramidal neurons. The dura mater adhered to V4 bilaterally and we were unable to perform histology. The second animal (Monkey A) is currently in use in another study. Thus, we do not have a measure of lenti/CAMKII selectivity in macaque V4. However, it is reasonable to assume that the opsin was strongly biased toward pyramidal neurons. The viral constructs and injection protocol used in the present study were identical to those used previously in V1 (*Nassi et al., 2015*). In that study, lenti/CAMKII expressing C1V1/EYFP led to expression that was highly selective for excitatory neurons (Figure 1B of *Nassi et al., 2015*): of 119 neurons imaged in five different fields of view, only 2 (1.7%) were double labeled for both EYFP and parvalbumin/calretinin/calbindin, indicating that expression was heavily biased toward excitatory neurons. CaMKII has also been used in other macaque brain areas including perirhinal cortex (*Tamura et al., 2017*), where it also led to opsin expression primarily in excitatory neurons. It is possible that lenti/CAMKII leads to less selective expression in V4 pyramidal neurons, but whatever the degree of selectivity we were able to activate neuronal responses in a phase-dependent manner with low intensity laser stimulation. as needed to test the effect of correlated variability on perceptual discrimination in the present study.

Our results establish the first causal link between correlated variability and perception. The optogenetic stimulation protocol in our study, using sinusoidal modulation of laser irradiance, induces the kind of correlations in a local population of the cortex that might not be physiologically realistic. It nevertheless establishes the causal relevance of low-frequency correlated variability in perception and supports the hypothesis that attention-dependent reductions in correlated variability enhance perception. Recently, studies have theorized that only certain correlations – those that are indistinguishable from stimulus-induced correlations – are information limiting (*Moreno-Bote et al., 2014*). We speculate that the correlations induced in our study included such information-limiting correlations, resulting in the observed impairment. The timescale of these low-frequency correlations is consistent with inter-saccadic intervals (200–300 ms), which may be a relevant timeframe for gathering visual information (*Yarbus et al., 1967*). Decreases in correlated variability at this timescale could therefore be critical for perception. Our study paves the way for investigating the laminar and cell-class specific components of the cortical circuit that determine this critical component of perception.

## Materials and methods

### Surgical procedures

Surgical procedures have been described in detail previously (*Nandy et al., 2017*; *Nassi et al., 2015*; *Ruiz et al., 2013*). In brief, an MRI compatible low-profile titanium chamber was placed over the pre-lunate gyrus, on the basis of preoperative MRI imaging in two rhesus macaques (right hemisphere in Monkey A, left hemisphere in Monkey C). The native dura mater was then removed and a silicone based optically clear artificial dura (AD) was inserted, resulting in an optical window over dorsal V4 (*Figure 1A,B*). All procedures were approved by the Institutional Animal Care and Use Committee and conformed to NIH guidelines.

### Viral injections

Viral injection procedures have been described in detail previously (*Nassi et al., 2015*). In brief, we injected a VSVg-pseudotyped lentivirus carrying the C1V1-EYFP gene behind the 1.3kb CaMKIIα promoter (lenti-CaMKIIa-C1V1$_{E162T}$-ts-EYFP; titer = $3 \times 10^{10}$ TU/ml) at 2 cortical sites in monkey A and 1 cortical site in monkey C while they were anesthetized and secured in a stereotactic frame. The viral constructs were chosen to preferentially drive expression of the depolarizing opsin C1V1 in excitatory neurons local to the injection site (*Han et al., 2009*). We injected approximately 0.5µl of virus at each depth in 200µm increments across the full 2mm thickness of cortex. All injections were targeted to para-foveal regions of V4 with eccentricities between 5 and 8 degrees of visual angle. Expression of the fluorescently tagged opsin was confirmed using epifluorescence goggles (BLS Ltd., Budapest, Hungary) after about 4-6 weeks of viral injection (*Figure 1C*).

## Opto-Electrophysiology

At the beginning of each recording session, a plastic insert, with an opening for targeting electrodes and for optical stimulation, was lowered into the chamber and secured. This served to stabilize the site against cardiac and respiratory pulsations. The opening was centered at the site of viral injection. A single tungsten microelectrode (FHC Inc) was mounted on an adjustable X-Y stage attached to the recording chamber and advanced into the injection site using a micromanipulator (Narishige Inc) until a spike (single neuron or multi-unit) could be reliably isolated from background voltage fluctuations. Site targeting was done under microscopic guidance (Zeiss Inc) using the microvasculature as reference. A single optical fiber (600 μm multimode fiber, 0.37NA, Thorlabs Inc) was mounted on the same X-Y stage and positioned over the injection site perpendicular to the calvarium. The microelectrode was positioned at an angle of 20 degrees with respect to the optical fiber (see schematic in *Figure 1A*).

We used a 532 nm diode-pumped solid-state (DPSS) laser (OEM Laser Systems Inc) as the light source for optical stimulation. The laser was placed on an optical breadboard in-line with a Uniblitz mechanical shutter (Vincent Associates), electro-optical modulator ('EOM', ConOptics Inc) and an optical fiber collimator/coupler (Thorlabs, Inc) attached to the optical fiber. A beam-splitter between the EOM and collimator directed approximately 1% of the light toward a high-speed photo-detector (Thorlabs, Inc). The EOM allowed us to control the intensity of laser light entering the fiber and was controlled using custom-written Labview software and a National Instruments digital acquisition board. Before each experiment we calibrated the output of the high-speed photodetector to the full range of intensities (irradiance units) measured at the fiber tip using an integrating sphere photodiode power sensor and a digital power meter (Thorlabs, Inc). This enabled real-time, calibrated irradiance measurements during all experiments.

Neuronal signals were recorded extracellularly, filtered, and stored using the Multichannel Acquisition Processor system (Plexon Inc). Neuronal signals were classified as either multi-unit clusters or isolated single units using Plexon Offline Sorter software. Single units were identified based on two criteria: (a) if they formed an identifiable cluster, separate from noise and other units, when projected into the principal components of waveforms recorded on that electrode and (b) if the inter-spike interval (ISI) distribution had a well defined refractory period.

Data was collected over 42 sessions (24 sessions in Monkey A, 18 in Monkey C), yielding a total of 94 units. Frequently, multiple units could be identified while recording from the single tungsten electrodes. Data was collected over an additional three sessions for control analyses (*Figure 2—figure supplement 4*).

## Task and stimuli

Stimuli were presented on a computer monitor placed 57 cm from the eye. Eye position was continuously monitored with an infrared eye tracking system (ISCAN ETL-200). Trials were aborted if eye position deviated more that 1° (degree of visual angle, 'dva') from fixation. Experimental control was handled by NIMH Cortex software (http://www.cortex.salk.edu/).

### Receptive Field (RF) Mapping

At the beginning of each recording session, neuronal RFs were mapped using subspace reverse correlation (*Ringach et al., 1997*) in which Gabor (eight orientations, 80% luminance contrast, spatial frequency 1.2 cycles/degree, Gaussian half-width 2°) or ring stimuli (80% luminance contrast) appeared at 60 Hz while monkeys maintained fixation. Each stimulus appeared at a random location selected from an 11 × 11 grid with 1° spacing in the appropriate visual quadrant. All RFs were in the lower visual quadrant (lower-left in Monkey A, lower-right in Monkey C) and with eccentricities between 5 and 8 dva.

### Irradiance response curves

After estimating the RF of a single-unit or multi-unit cluster, we assessed its sensitivity to optical stimulation. While the monkey maintained fixation, we measured the neuronal response to visual (achromatic Gabor stimulus, spatial frequency 1.2 cycles/degree, 20% luminance contrast) and optical stimulation. The visual stimulus was flashed at the RF for 200 ms with a simultaneous step laser

pulse chosen from one of several irradiance values (typically 0, 10, 30, 50 and 70 mW/mm$^2$) (*Figure 2—figure supplement 5*).

## Attention task

In the main experiment, monkeys had to perform an attention-demanding orientation change-detection task (*Figure 2A*). While the monkey maintained fixation, two achromatic Gabor stimuli (spatial frequency 1.2 cycles/degree, 6 contrasts randomly chosen from an uniform distribution of luminance contrasts, $c = [10, 18, 26, 34, 42, 50\%]$) were flashed on for 200ms and off for a variable period chosen from a uniform distribution between 200-400ms. One of the Gabors was flashed in the center of the RFs, the other at a location of equal eccentricity across the vertical meridian. At the beginning of a block of trials, the monkey was spatially cued ('instruction trials') to covertly attend to one of these two spatial locations. During these instruction trials, the stimuli were only flashed at the spatially cued location. At an unpredictable time (minimum 1s, maximum 5s, mean 3s), one of the two stimuli changed in orientation. The time of orientation change was chosen by sampling from an exponential distribution (thus leading to a flat hazard function of wait times till orientation change). If the sampled change time exceeded 5s, the trial was treated as a catch trial (see below), in which the change did not actually occur during the trial and the monkey was rewarded for maintaining fixation. If the orientation change did occur, the monkey was rewarded for making a saccade to the location of orientation change. However, the monkey was rewarded for only those saccades where the saccade onset time was within a window of 100-400ms after the onset of the orientation change. The location of orientation change was chosen with 95% probability at the cued location and with 5% probability at the uncued location ('foil trials'). We controlled task difficulty by varying the degree of orientation change ($\Delta_{ori}$), which was randomly chosen from one of 8 orientations in the range 1-15°. The orientation change in the foil trials was fixed at 4°. These foil trials allowed us to assess the extent to which the monkey was using the spatial cue, with the expectation that there would be an impairment in performance and slower reaction times (*Figure 2—figure supplement 2A*) compared to the case in which the change occurred at the cued location. If no change occurred before 5s, the monkey was rewarded for maintaining fixation ('catch trials', 13% of trials). We will refer to all stimuli at the baseline orientation as 'non-targets' and the stimulus flash with the orientation change as the 'target'. If the monkey made a saccade to a non-target stimulus at any time, it was treated as a false alarm and the monkey was not rewarded.

On a random subset of trials (50% of trials in experimental sessions with low-frequency stimulation only; 33% of trials in experimental sessions with both low- and high-frequency stimulation conditions), neurons at the injection site were stimulated with 4–5 Hz sinusoidally modulated low-power laser stimulation ('low-frequency stimulation' condition). The sinusoidal modulation had excursions from a minimum irradiance close to 0 mW/mm$^2$ to a maximum irradiance, chosen such that the equivalent root-mean-squared intensity elicited a firing rate either 10% above (*Figure 2—figure supplement 5*, left example unit) or 10% below (*Figure 2—figure supplement 5*, right example unit) the firing rate in the zero-irradiance condition. The optical stimulation lasted the entire duration of the trial. On a subset of experimental sessions (n = 15), neurons at the injection site were also stimulated with 20 Hz sinusoidally modulated low-power laser stimulation ('high-frequency stimulation' condition; 33% of trials).

## Data analysis

### Behavioral Analysis

For each orientation change condition $\Delta_{ori}$, we calculated the hit rate as the ratio of the number of trials in which the monkey correctly identified the target with a saccade over the number of trials in which the target was presented. The hit rate as a function of $\Delta_{ori}$, yields a behavioral psychometric function (*Figure 2B*, *Figure 2—figure supplement 1*, *Figure 2—figure supplement 2*). Psychometric functions were fitted with a smooth logistic function (Palamedes MATLAB toolbox). Error bars were obtained by a jackknife procedure (20 jackknives, 5% of trials left out for each jackknife). Performance for the foil trials were calculated similarly as the hit rate for trials in which the orientation change occurred at the un-cued location (*Figure 2—figure supplement 2A*, left panel, square symbol). Performance for the catch trials was calculated as the fraction of trials in which the monkey correctly held fixation for trials in which there was no orientation change (*Figure 2—figure supplement*

*2A*, left panel, star symbol). Psychometric functions were obtained separately for the baseline (no laser stimulation) and the optical stimulation conditions.

Psychophysical studies have found that human observers are better able to discriminate stimulus orientations near the cardinal than oblique orientations (*Girshick et al., 2011*; *Heeley and Timney, 1988*; *Appelle, 1972*; *Orban et al., 1984*; *Campbell et al., 1966*). Electrophysiological and imaging studies in humans, monkeys, cats and ferrets have found that cardinal orientations are overrepresented in V1 (*De Valois et al., 1982*; *Furmanski and Engel, 2000*; *Li et al., 2003*; *Wang et al., 2003*; *Chapman and Bonhoeffer, 1998*). Consistent with this, we find that monkeys performed the task better during sessions in which they were required to discriminate orientation changes from cardinal (0°, 90°) non-target orientations, as reflected in elevated performance in detecting the smallest orientation change ($p = 0.05$, t-test) and elevated threshold ($p = 0.002$, t-test) for cardinal compared to non-cardinal orientations (*Figure 2—figure supplement 2D*). Threshold is the stimulus condition at which performance was mid-way between the lower and upper asymptotes of the fitted psychometric function.

We characterized the change in behavioral performance due to optical stimulation as a modulation index (PDMI, perceptual discrimination modulation index):

$$\mathrm{PDMI} = \left( \left( threshold_{opto} - threshold_{baseline} \right) / threshold_{baseline} \right)$$

In addition, we assessed any changes in psychometric function slope (steepness of the curve at threshold) due to optical stimulation as the change over the baseline (no laser) condition normalized by the slope at baseline (*Figure 2—figure supplement 3A*). Significant changes in threshold and slope for each individual behavioral session (*Figure 2—figure supplement 3B*) were calculated by comparing the distributions of threshold and slope values estimated from the jackknife procedure between the optical stimulation and baseline conditions (t-test, $\alpha = 0.05$, corrected for the number of jackknifes).

## Peri-stimulus time-histograms

For this and subsequent analyses of neuronal data, we restricted our analyses to non-target flashes from correct trials (hit trials in which the monkey correctly detected a target or correct catch trials). Neuronal responses were binned using a sliding window of width 30 ms that was shifted by 10 ms increments to obtain the time-varying firing rates, also known as the peri-stimulus time-histograms (PSTH), of the recorded units (*Figure 4A*). Population PSTH plots (*Figure 4B*) were obtained after normalizing the responses of each neuron to the peak across the four experimental conditions (two attention conditions [attend-in, attend-away] x two stimulation conditions [no stimulation, laser stimulation]).

## Spike-phase distributions

We calculated the phase of each spike with respect to the sinusoidal laser stimulation during a 200 ms blank period before a non-target stimulus flash. We only considered those inter-stimulus periods where the inter-stimulus interval was greater than 500 ms (in other words, the interval between onset of the stimulus and the offset of the previous stimulus was greater than 300 ms), so as to minimize artifacts due to stimulus offset. Polar plots in *Figure 4C* show the distributions of spiking phases. To see if these distributions were significantly different from chance, we calculated a null distribution by generating spike times from a rate-matched Poisson process (gray polar plots in *Figure 4C*). To obtain reliable estimates for spike-phase distributions, we restricted our analysis to units with a minimum firing rate of 5 spikes/s (n = 68; firing rate averaged over the 200 ms stimulus-evoked period between 60–260 ms after non-target onset).

## Spike-count correlations ($r_{SC}$)

We calculated the Pearson correlation of spike counts across trials for every pair of simultaneously recorded units. In order to remove the influence of confounding variables like stimulus strength, spike counts were z-scored using the mean and standard deviation for repetitions of each stimulus type. Ordered pairs of z-scored spike counts were collapsed across contrast conditions and the Pearson correlation was calculated from these ordered pairs. This was done separately for the different attention and optical stimulation conditions and also for different sized counting windows (50ms for

high-frequency correlations, 200ms for low-frequency correlations) during the stimulus-evoked period between 60-260ms after non-target onset (*Figure 3*). Multiple non-overlapping windows were used for those counting windows that were smaller than the 200ms stimulus evoked period.

## Neural Discrimination Modulation Index (NDMI)

For each neuron in our population, we extracted spike counts to repeated presentations of non-target and target stimuli (60-260ms after stimulus onset) for the baseline and low-frequency stimulation conditions when the animals were attending in to the RF. Response rates for each neuron were normalized by the maximum response across conditions after first subtracting the spike rates during a 200ms pre-stimulus period (pre-stimulus rates calculated separately for baseline and low-frequency stimulation conditions). We thus obtained two response clouds for each experimental condition: one for the non-target stimuli and the other for the target stimuli (schematic in *Figure 5A*). We calculated the neural discriminability between the two response clouds as the Mahalanobis distance ($D$) between the centroid of the target responses and the non-target response cloud. The modulation of this discriminability measure due to optical stimulation was quantified as an index (Neural Discriminability Modulation Index, NDMI):

$$\text{NDMI} = \frac{D_{opto} - D_{baseline}}{D_{opto} + D_{baseline}}$$

NDMI for each experimental session was calculated in two ways: either from multi-dimensional clouds from all simultaneously recorded neurons or as the average of two-dimensional clouds across all pairs of simultaneously recorded neurons.

## Computational model

A similar model has been described previously (*Nandy et al., 2017*). We set up a conductance-based model of $N_E$ excitatory and $N_I$ inhibitory neurons with 80% connection probability (both within and across the two populations) and with the following synaptic weights (*Figure 6*):

$$\text{E to E: } w_{EE} = \frac{W_{EE}}{N_E}; \; \text{I to I: } w_{II} = \frac{W_{II}}{N_I}; \; \text{E to I: } w_{IE} = \frac{W_{IE}}{N_E}; \; \text{I to E: } w_{EI} = \frac{W_{EI}}{N_I}$$

We simulated models of $N_E = 800$ excitatory and $N_I = 200$ inhibitory spiking units. The spiking units were modeled as Izhikevich neurons (*Izhikevich, 2003*) with the following dynamics:

$$\frac{dv}{dt} = 0.04v^2 + 5v + 140 - u + I$$

$$\frac{du}{dt} = a(bv - u)$$

$$\text{if } v = 30mV, \text{ then } v \leftarrow c \text{ and } u \leftarrow u + d$$

$v$ is the membrane potential of the neuron and $u$ is a membrane recovery variable. $I$ is the current input to the neuron (synaptic and injected DC currents). The parameters $a$, $b$, $c$ and $d$ determine intrinsic firing patterns and were chosen as follows:

$$\text{Excitatory units: } a = 0.02, \; b = 0.2, \; c = -65, \; d = 8$$

$$\text{Inhibitory units: } a = 0.1, \; b = 0.2, \; c = -65, \; d = 2$$

Presynaptic spikes from excitatory units generated fast (AMPA) and slow (NMDA) synaptic currents, while presynaptic spikes from inhibitory units generated fast GABA currents:

$$I_{syn} = \sum_i g_{AMPA}(t)(v(t) - V_{AMPA}) + \sum_j g_{NMDA}(t)(v(t) - V_{NMDA}) + \sum_k g_{GABA}(t)(v(t) - V_{GABA})$$

where $V_{AMPA} = 0, V_{NMDA} = 0, V_{GABA} = -70$ are the respective reversal potentials (mV). The synaptic time courses $g(t)$ were modeled as a difference of exponentials:

$g(t) = \frac{1}{\tau_d - \tau_r}\left[exp\left(-\frac{t-\tau_l}{\tau_d}\right) - exp\left(-\frac{t-\tau_l}{\tau_r}\right)\right]$ where $\tau_l, \tau_r$ and $\tau_d$ are the latency, rise and decay time constants with the following parameter values (ms) (*Brunel and Wang, 2003*):

|  | $\tau_l$ | $\tau_r$ | $\tau_d$ |
|---|---|---|---|
| AMPA | 1 | 0.5 | 2 |
| NMDA | 1 | 2 | 80 |
| GABA | 1 | 0.5 | 5 |

The NMDA to AMPA ratio was chosen as 0.45 (*Myme et al., 2003*). The network was stimulated by a DC step current ($I_E = 2.8$, $I_I = 2.3$) of duration 1.5s (*Figure 6B*). Synaptic noise was simulated by drawing from a normal distribution ($I_{syn-noise} \sim N(\mu = 0, \sigma = 3)$). To simulate the laser stimulation in the main experiment, we chose a random subset (50%) of excitatory units to which we injected a 4Hz sinusoidally modulated current ($I_{opto}$; meancurrent = 0.5; peaktotroughrange = 0.7). Such a current by itself did not produce spiking activity in the network.

We computed the spike-spike coherence between all pairs of excitatory units in the model (irrespective of whether the units were subjected to the additional sinusoidally modulated current) using multi-taper methods (*Mitra and Pesaran, 1999*), over a 400ms window for both simulation conditions: with and without $I_{opto}$. Spike trains were tapered with a single Slepian taper, giving an effective smoothing of 2.5Hz for the 400ms window (NW=1, K=1). To control for differences in firing rate between the two conditions, we adopted a rate matching procedure similar to (*Mitchell et al., 2009*). Induction of coherent activity in the network due to sub-threshold sinusoidal stimulation was calculated as a modulation index of coherence across the two conditions: $SSCMI = (SSC_{with} - SSC_{without})/(SSC_{with} + SSC_{without})$. In order to obtain a baseline for the coherence expected solely due to trends in firing time-locked to network stimulation, we also computed coherence in which trial identities were randomly shuffled (*Figure 6—figure supplement 1C-D*).

## Acknowledgements

This research was supported by NIH R01 EY021827 to JHR and ASN, the Gatsby Charitable Foundation to JHR, fellowship from the NIH T32 EY020503 training grant and the NARSAD Young Investigator Grant to ASN, The Salk Institute Excellerators Fellowship Program and the NARSAD Young Investigator Grant to JJN, NIH R00 EY025026 Pathways to Independence Award to MPJ, and by a NEI core grant for vision research P30 EY019005 to the Salk Institute. We would like to thank Ed Callaway and Euiseok Kim for help with optogenetic reagents, Rob Teeuwen for assistance with animal training and Catherine Williams and Mat LeBlanc for excellent animal care.

## Additional information

### Funding

| Funder | Grant reference number | Author |
|---|---|---|
| Brain and Behavior Research Foundation | | Anirvan Nandy Jonathan J Nassi |
| National Institutes of Health | R01 EY021827 | John Reynolds Anirvan Nandy |
| National Institutes of Health | NIH T32 EY020503 | Anirvan Nandy |
| National Institutes of Health | R00EY025026 | Monika P Jadi |
| NIH Blueprint for Neuroscience Research | | John Reynolds |
| Gatsby Charitable Foundation | | John Reynolds |

The funders had no role in study design, data collection and interpretation, or the decision to submit the work for publication.

## Author contributions

Anirvan Nandy, Conceptualization, Data curation, Software, Formal analysis, Funding acquisition, Investigation, Methodology, Writing—original draft, Writing—review and editing, Designed the experiments, Collected and analyzed the data, Developed the computational model, Ran the simulations; Jonathan J Nassi, Conceptualization, Investigation, Designed the experiments; Monika P Jadi, Formal analysis, Visualization, Writing—review and editing; John Reynolds, Conceptualization, Supervision, Writing—original draft, Writing—review and editing, Designed the experiments, Wrote the manuscript

## Author ORCIDs

Anirvan Nandy (iD) https://orcid.org/0000-0002-4225-5349

## Ethics

Animal experimentation: This study was performed in strict accordance with the recommendations in the Guide for the Care and Use of Laboratory Animals of the National Institutes of Health. All of the animals were handled according to approved institutional animal care and use committee (IACUC) protocols of the Salk Institute. All procedures were approved by the Institutional Animal Care and Use Committee at the Salk Institute (Protocol #14-00014) and conformed to NIH guidelines.

## Decision letter and Author response

Decision letter https://doi.org/10.7554/eLife.35123.024
Author response https://doi.org/10.7554/eLife.35123.025

# Additional files

## Supplementary files

• Transparent reporting form
DOI: https://doi.org/10.7554/eLife.35123.020

## Data availability

Data for the main figures are available via Dryad (doi:10.5061/dryad.8v0k1j3).

The following dataset was generated:

| Author(s) | Year | Dataset title | Dataset URL | Database and Identifier |
|-----------|------|---------------|-------------|-------------------------|
| Nandy A, Nassi J, Jadi M, Reynolds J | 2019 | Data from: Optogenetically induced low-frequency correlations impair perception | http://dx.doi.org/10.5061/dryad.8v0k1j3 | Dryad Digital Repository, 10.5061/dryad.8v0k1j3 |

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
