## [Decision Letter]

Thank you for submitting your article "Optogenetically induced low-frequency correlations impair perception" for consideration by *eLife*. Your article has been reviewed by two peer reviewers, and the evaluation has been overseen by a Reviewing Editor and a Senior Editor. The reviewers have opted to remain anonymous.

The reviewers have discussed the reviews with one another and the Reviewing Editor has drafted this decision to help you prepare a revised submission.

Summary:

The paper examined whether optogenetically induced increases in spike count correlations in V4 neurons affect the ability of macaque monkeys to detect small orientation changes of Gabor gratings. The authors report that the optogenetic stimulation in the low but not high frequency range affected performance and this effect was limited to the attended location, concluding that the attention induced reduction of low frequency spike count correlations results in improved performance.

The reviewers commented on the ambitious nature of the work and its potential importance, but raised a number of serious reservations that must be addressed for the paper to be further considered for publication in *eLife*. These are summarized below.

Essential revisions:

1) Anatomical evidence confirming that the neurons were actually transfected should be provided. This includes the information about the affected cell type, their layer locations and variability of the expression.

2) Variable performance on the orientation change task makes it difficult to interpret the effects of optogenetic stimulation. This problem should be addressed.

3) The presence of on- and off- transients in visual stimulation occurring at a frequency of 3-5Hz creates a potential problem in the study aimed at detecting correlations occurring at similarly low frequency. It is important to rule out that optogenetic stimulation may be affecting the detection of the onsets and offsets of visual stimuli. Reviewer 1 suggests a control condition that would eliminate such transients to see whether optogenetic manipulation is still selective for low frequencies.

A related question (reviewer 2) concerns the phase of optical stimulation relative to stimulus presentation whether the presence of the behavioral deficits depended on the phase of optical stimulation. Also, provide information about the phase of optical stimulation used to compute noise correlations

4) Data presentation limited to averages and distributions does not allow the evaluation of significant effects in each experiment. The data suggest that in some cases optogenetic stimulation resulted in elevated thresholds and slopes and in some cases in opposite effects. This is a problem that needs to be addressed and discussed.

5) Please provide a direct comparison between firing rates with and without optical stimulation, showing effects on individual neurons. Rate modulation index does not allow the reader to assess such effects directly.

6) Was orientation tuning affected by optical stimulation? If they were, the disruption of orientation representation in V4 by stimulation could potentially explain the behavioral effects. This should be addressed.

7) Please explain cued and uncued locations vs. "catch trials" and their randomization during the experiment (reviewer 1).

*Reviewer #1:*

The goal of this paper is to causally test the idea that low-frequency oscillations amongst cortical neurons are a major limiting factor in visual perception and are actively controlled for visual attention. The strategy is a bold one – use optogenetics to introduce low-frequency correlations into the activity of cortical neurons in extrastriate area V4 and document the effects on both visual perception and neuronal activity. If the idea is correct, then the introduction of low-frequency correlations should impair performance. But there are many reasons that performance might be impaired when you alter visual cortical activity. So it is also necessary to show that visual activity has not simply been disrupted by the optogenetic manipulation, and that it is specifically the temporal restructuring of the activity that is crucial. To this end, the paper shows that overall firing rates are not changed, and that the effects are found only with low-frequency and not high-frequency stimulation, among other controls.

This is an ambitious set of experiments to attempt to pack into a short paper, especially given all of the potential pitfalls and controls that need to be considered. In the end, I was not fully convinced that the results as presented support the conclusions. There are several basic issues that need to be addressed to make the case more convincing. I also think that a more detailed unpacking of the data is need in order to understand the results.

The most obvious missing piece is histology. Figure 1C shows a surface view of the cortex illustrating the expression of EYFP. This is not sufficient to answer the questions relevant for interpreting the behavioral and neuronal data. Can you confirm that neurons were transfected? Which types of neurons, and in which layers? How variable was the expression across layers? Can you rule out retrograde transfection from axon terminals? Without histology to verify what was stimulated, I find it difficult to interpret the results.

Performance in the orientation change task seems extremely variable, to the point that it raises concerns about how to interpret changes in task performance. In some cases, the thresholds for detecting orientation changes seem to be in the expected range (a few degrees) – for example, Figure 2B and Figure 2—figure supplement 2A. But in other cases, the thresholds are unusually high, 10 degrees or more (Figure 2—figure supplement 2C). There are also unexpected elevations in% correct for signal values that should be below threshold – for example, 40% correct for the lowest orientation change value in Figure 2—figure supplement 2B. How could the monkey get 40% correct for an undetectable signal? In order to interpret the optogenetic manipulation, I would need to be reassured about the reliability of task performance in the absence of optogenetic manipulation.

There is a very basic aspect of the experimental design that seems like a problem, but perhaps the authors have a very well-reasoned explanation for this approach. The hypothesis is that low-frequency (4-5 Hz) correlations play a central role in cortical processing, and this guides the choice of sinusoidal optogenetic stimulation. But the stimulus itself is flashed on for 200 ms and then left off for 200-400 ms, which means that there were visual transients (on or off) also occurring at a frequency of 3-5 Hz. If the goal is to test the importance of intrinsic low-frequency correlations, why use a stimulus that includes transients in this same frequency range? It seems that an alternative explanation for the perturbations in performance might be that the optogenetic stimulation masks the ability to detect the onsets and offsets of the visual stimulus. Can this be ruled out? If the stimulus were simply left on for an extended period of time (seconds), and then changed orientation, would the optogenetic manipulation still be effective and still selective for low frequencies?

The data were compressed for presentation in ways that made it difficult to understand what additional significant effects might be present in each experiment. Figure 2C is a good example of this. The histogram illustrates that, overall, both the threshold and slope increased in trials with low-frequency optical stimulation, by showing that the population-level ratio of stimulation relative to baseline is significantly greater than 1. First, please clarify the stats (here and elsewhere). This is reported as a t-test in the fourth paragraph of the Results and Discussion. Which type – paired sample? Is the distribution normal, or should a non-parametric test be used? Second, please report the data in a way that allows us to appreciate what happened on individual experiments. Given the scale on the x-axis, it looks like optical stimulation may have caused significant increases in threshold or slope in some experiments and significant decreases in others. If so, it would be misleading to base the conclusion on the average effect. This point is especially relevant for the attend away condition (Figure 2D), in which the average threshold is not different, but the spread in the histogram suggests that many individual experiments were significant but roughly equally split between increases and decreases.

I had similar concerns about the neuronal data in Figure 4. In addition to plotting the rate modulation index, please directly compare the firing rates with and without optical stimulation. Was the firing rate significantly changed for individual neurons? The effects of attend-in and attend-away should be similarly documented across the population of neurons. Did optical stimulation significantly change this modulation for individual neurons? The concern here is that there may be no difference on average, but a more or less balanced combination of significant increases and significant decreases.

Beyond changes in firing rate, it is also possible that the optical stimulation disrupted the tuning properties of the neurons. Do you have data to confirm that the orientation tuning and receptive field properties of the neurons was not changed during optical stimulation? If the claim is specifically about the role of low-frequency correlations, it is important to rule out the possibility that behavioral effects were due simply to disrupting the representation of orientation information in V4.

*Reviewer #2:*

Nandy and colleagues investigate whether optogenetically induced increases in spike count correlations in V4 neurons affect the ability of macaque monkeys to detect small orientation changes of Gabor gratings. The find that this is indeed the case, but only if the optogenetic stimulation is in the low frequency range (4-5Hz, sinusoidal modulation), not when it occurs at higher frequencies (tested were 20 Hz). This only occurred when the neurons stimulated represented the attended (instructed) location, not when the represented the non-instructed location. The authors conclude that the attention induced reduction of low frequency spike count correlations indeed convey behavioural benefits.

This is an interesting study, but I have a few questions:

I was unable to determine whether the phase of stimulation was fixed relative to stimulus presentation. Probably not as stimulus presentation varied within a trial. This means that orientation changes would occur at variable phases of optogenetic stimulation. If so, it will be important to know whether behavioural deficits occurred equally across different phases, or whether they were co-modulated.

The histograms show and the noise correlations also seem to be calculated over different optogenetic phases (if my assumption from above is correct). It will be important to determine whether there were changes in either when calculated for different optogenetic phases. If so, these need to be documented in detail.

The authors state that no effect was seen in the attend away condition. Was that even true on catch trials (i.e. attend away, but change occurs at RF)? If so, it needs to be explained why.

Why was the higher frequency in the beta range, not in the gamma range, where some people might expect to see behavioural benefits to occur?

[Editors' note: further revisions were requested prior to acceptance, as described below.]

Thank you for submitting your article "Optogenetically induced low-frequency correlations impair perception" for consideration by *eLife*. Your article has been reviewed by two peer reviewers, and the evaluation has been overseen by a Reviewing Editor and Eve Marder as the Senior Editor. The reviewers have opted to remain anonymous.

The reviewers have discussed the reviews with one another and the Reviewing Editor has drafted this decision to help you prepare a revised submission.

Summary:

The reviewers acknowledged your efforts to address the comment raised in their initial review. However, a number of issues remain and those undermine their confidence in the main conclusions of the paper. These must be addressed for the manuscript to be considered for publication in *eLife*. The points listed below are based not only on the reviews attached below, but also on the discussion among the reviewers that followed.

Essential revisions:

1) Lack of histology

The reviewers felt that the lack of histology seriously complicates the interpretation of the results but are willing to accept a possibility that the expression patterns of the viral vector in V4 are the same as those shown for V1 in the previous publication from the lab. However, you should add appropriate disclaimers about the assumptions about transfected neurons (based on the published V1 data), and discuss how your interpretation would be affected if these assumptions turn out to be incorrect.

2) Variability in orientation thresholds

The reviewers felt that the explanation attributing variability to the difference in thresholds for oblique and cardinal orientation, needs stronger documentation. To that end, rather than providing 8 example psychometric functions, the relationship between base orientation and performance should be documented by plotting thresholds for each of the sessions as a function of orientation.

3) Potential confound of visual and optogenetic stimulation delivered at similar low frequencies

Reviewers did not feel that the phase analysis included in the revision adequately addressed the problem. Reviewer 2 commented that while the overall effect may not be phase-specific, it may be specific "to cases where the optogenetic frequency is similar to the stimulus onset/offset frequency". Please provide additional information, to address this.

Please address the absence of label for the x-axis in Figure 4—figure supplement 3B and different sign of effects on threshold and sensitivity. One of the reviewers suggests sorting the experiments into subsets based on how the psychometric curves change (for example, based on where the experiments fall in Figure 2—figure supplement 5).

4) Changes in thresholds and slopes

The summary plot (Figure 2—figure supplement 5) reveals not only increases in threshold (22/42 cases) but also decreases in threshold (~14 cases). Please address the apparently opposite effects produced by stimulation.

In addition, it appears that stimulation increased the slope more often than decrease it. Since the increase in slope is likely to be associated with an increase in sensitivity, the proposed role of decreased correlations in improving the separability of neuronal activity is puzzling. This effect appears inconsistent with the proposed interpretation of the changes in spike-count correlations with stimulation. The reviewers suggest that you link the measured changes in correlations with the observed changes in psychometric functions by using a decoding scheme similar to that used by Cohen and Maunsell, 2009. This approach would allow you to predict the changes in psychometric curves given the changes in spike-count correlations.

5) Orientation tuning

Please provide additional information concerning the Figure 4—figure supplement 4 scaling on the x-axis and pooling data across neurons. How were the neurons used for this analysis selected? (see comment from reviewer 1).

6) Please provide documentation of false alarm rates from catch trials with and without optogenetic stimulation (see comment from reviewer 1).

7) The phase locking of spikes to the phase of stimulation should also be shown during the stimulus, in addition in the absence of stimulation.

8) The effect of optogenetic phase on behavior should be aligned with the preferential population activity, which seems to peak/trough at 240° and 60° respectively.

9) Please clarify what statistical test was used to determine that the phase of optical stimulation did not affect behavior or orientation tuning

*Reviewer #1:*

In my comments below, for clarity I repeat the original comments in quotes, with my new comments inserted after each item.

Overall, the authors have made efforts to address each major comment, but have been prevented from fully settling several of the points due to technical limitations. Some of these are substantial points that affect my confidence in the main conclusions.

"The most obvious missing piece is histology. Figure 1C shows a surface view of the cortex illustrating the expression of EYFP. This is not sufficient to answer the questions relevant for interpreting the behavioral and neuronal data. Can you confirm that neurons were transfected? Which types of neurons, and in which layers? How variable was the expression across layers? Can you rule out retrograde transfection from axon terminals? Without histology to verify what was stimulated, I find it difficult to interpret the results."

The authors do not have histology for the two monkeys used in the study but they show a figure from a previously published study using the same viral vector injected into cortical area V1.

Will the expression patterns be the same in V4, the area targeted in this study? I don't know, and I don't know of any published work using this vector in macaque V4. Apparently, there is tissue from one of the monkeys (that was sacrificed) from this study; even if the tissue is damaged due to penetrations it should be possible to identify transfected cells, and the approximate size and layer distribution. It is not clear what steps were taken along these lines, or if that tissue was unfortunately discarded.

"Performance in the orientation change task seems extremely variable, to the point that it raises concerns about how to interpret changes in task performance. In some cases, the thresholds for detecting orientation changes seem to be in the expected range (a few degrees) – for example, Figure 2B and Figure 2—figure supplement 2A. But in other cases, the thresholds are unusually high, 10 degrees or more (Figure 2—figure supplement 2C). There are also unexpected elevations in% correct for signal values that should be below threshold – for example, 40% correct for the lowest orientation change value in Figure 2—figure supplement 2B. How could the monkey get 40% correct for an undetectable signal? In order to interpret the optogenetic manipulation, I would need to be reassured about the reliability of task performance in the absence of optogenetic manipulation."

The authors explain that this is probably due to differences in detection performance – in particular, thresholds were lower for baseline orientations near the cardinal (i.e., horizontal and vertical) orientations. They show 8 sample psychometric curves that are consistent with this explanation.

This seems plausible, but the author should show it holds true for the other 34 sessions as well. If you simply plot threshold for each session as a function of the baseline orientation, this would show whether the variance was indeed systematically related to the baseline orientation.

"There is a very basic aspect of the experimental design that seems like a problem, but perhaps the authors have a very well-reasoned explanation for this approach. The hypothesis is that low-frequency (4-5 Hz) correlations play a central role in cortical processing, and this guides the choice of sinusoidal optogenetic stimulation. But the stimulus itself is flashed on for 200 ms and then left off for 200-400 ms, which means that there were visual transients (on or off) also occurring at a frequency of 3-5 Hz. If the goal is to test the importance of intrinsic low-frequency correlations, why use a stimulus that includes transients in this same frequency range? It seems that an alternative explanation for the perturbations in performance might be that the optogenetic stimulation masks the ability to detect the onsets and offsets of the visual stimulus. Can this be ruled out? If the stimulus were simply left on for an extended period of time (seconds), and then changed orientation, would the optogenetic manipulation still be effective and still selective for low frequencies?"

The authors agree these are important potential confounds but for technical reasons, they are not able to do the control experiment in which the stimulus is simply left on, to test whether the optogenetic stimulus might be masking the visual onset and offsets that occur in the same frequency range.

The phase analysis is interesting, but does not address the same point exactly. I wouldn't necessarily expect the effect to be phase-specific, but I do suspect it might be specific to cases where the opto frequency is similar to the stimulus onset/offset frequency. This question remains open.

Some comments about Figure 4—figure supplement 3B: What is the x-axis and why is it unlabeled? There seem to be some interesting possible mixed effects at low delta orientations. If you pool across all sessions (with different sign of effects on threshold and sensitivity) perhaps some effects are getting averaged out. Have you tried sorting the experiments into subsets based on how the psychometric curves change (for example, based on where the experiments fall in Figure 2—figure supplement 5)?

"The data were compressed for presentation in ways that made it difficult to understand what additional significant effects might be present in each experiment. Figure 2C is a good example of this. The histogram illustrates that, overall, both the threshold and slope increased in trials with low-frequency optical stimulation, by showing that the population-level ratio of stimulation relative to baseline is significantly greater than 1. First, please clarify the stats (here and elsewhere). This is reported as a t-test in the fourth paragraph of the Results and Discussion. Which type – paired sample? Is the distribution normal, or should a non-parametric test be used? Second, please report the data in a way that allows us to appreciate what happened on individual experiments. Given the scale on the x-axis, it looks like optical stimulation may have caused significant increases in threshold or slope in some experiments and significant decreases in others. If so, it would be misleading to base the conclusion on the average effect. This point is especially relevant for the attend away condition (Figure 2D), in which the average threshold is not different, but the spread in the histogram suggests that many individual experiments were significant but roughly equally split between increases and decreases."

The authors now provide a summary plot (Figure 2—figure supplement 5) that summarizes the changes in threshold and slope. This is helpful. It shows that in addition to the main effect reported in the paper – the increase in threshold seen in 22/42 cases – there is also sometimes a significant decrease in threshold (~14 cases). Any ideas about why the effect flips sign in these cases?

More curiously, and harder to understand, the stimulation also tends to increase the slopes more often than it significantly decreases the slopes. An increase in slope would imply that the sensitivity of the monkey during the stimulation had increased. Given the proposed role of decreased correlations in improving the separability of neuronal activity, shouldn't the main effect have been a decrease in slope?

"I had similar concerns about the neuronal data in Figure 4. In addition to plotting the rate modulation index, please directly compare the firing rates with and without optical stimulation. Was the firing rate significantly changed for individual neurons? The effects of attend-in and attend-away should be similarly documented across the population of neurons. Did optical stimulation significantly change this modulation for individual neurons? The concern here is that there may be no difference on average, but a more or less balanced combination of significant increases and significant decreases."

The authors have added a Figure 4—figure supplement 1 that compares the spike rates for all neurons with and without optogenetic stimulation, and report no significant change in firing rate due to stimulation. I find this set of figures very convincing.

"Beyond changes in firing rate, it is also possible that the optical stimulation disrupted the tuning properties of the neurons. Do you have data to confirm that the orientation tuning and receptive field properties of the neurons was not changed during optical stimulation? If the claim is specifically about the role of low-frequency correlations, it is important to rule out the possibility that behavioral effects were due simply to disrupting the representation of orientation information in V4."

The authors respond that they did not measure tuning curves. However, they do have data from the non-target orientation and some target orientations, which they report in Figure 4—figure supplement 4. I find it difficult to evaluate this plot because I don't understand the scaling on the x-axis or how data were pooled across neurons. It is also not clear that data from all neurons should be included in this analysis, unless their activity was strongly modulated across the range of orientations used (i.e., the data indicate direction tuning over the domain tested). And then the data might be aligned on the x-axis so that 1 value corresponded to the "best" direction.

The issue of possible changes in neuronal tuning is critical for interpreting the results. Perhaps the authors can do more to address this.

"There was some ambiguity in the description of how the orientation change in the stimulus was managed. The paper describes a 95% probability at the cued location and 5% at the uncued location. But then there were also "catch trials" without any change. It's not clear how these add up. Were the 95% and 5% independent? What was the probability of a catch trial? Were these truly randomized, or were they presented in as a fixed fraction of the trials?"

The authors now explain their definition of 'catch' trials.

Are the FA rates on from catch trials documented somewhere in the paper? I did not see it except for Figure 2—Figure supplement 2A, which curiously appears to show a false alarm rate on catch trials of about 50% Is this correct? How can the FA rate be that high when the hit rate drops well below that for small orientation changes? I would expect the FA rate to be the floor for the curve.

Aside from trying to understand the plots, the other reason for asking about FAs is to know whether the FAs also changed with optogenetic stimulation. This would be important for assessing possible changes in response criterion, which would also be important to nail down, since changes in criterion could also shift the psychometric curves

*Reviewer #2:*

The authors have addressed some of my previous points, but I have a few issues remaining:

They describe the phase locking of spikes to the phase of stimulation when no stimulus was present. However, it would be important to see this also for the stimulus period.

The authors use 4 bins to calculate the effect of optogenetic phase on behaviour, but these are not aligned with the preferential population activity alignment, which seems to peak/trough at 240° and 60° respectively. This needs to be done.

It is unclear what statistical test was used to determine whether behaviour was unaffected by the phase of optical stimulation?

The same is true for the effect on orientation tuning.

In general, statistical reporting should be checked and adequately improved.

[Editors' note: further revisions were requested prior to acceptance, as described below.]

Thank you for submitting your article "Optogenetically induced low-frequency correlations impair perception" for consideration by *eLife*. Your article has been reviewed by two peer reviewers, and the evaluation has been overseen by a Reviewing Editor and Eve Marder as the Senior Editor. The reviewers have opted to remain anonymous.

The reviewers have discussed the reviews with one another and the Reviewing Editor has drafted this decision to help you prepare a revised submission.

Summary:

The revised manuscript has addressed most of the reservations raised by the two reviewers. There remain only a few issues that need the authors attention. These are listed below.

Essential revisions:

1) Please substitute multiple t-tests with ANOVA to reveal potential interactions.

2) Please address the point concerning correction analysis, raised by reviewer 2.

3) Please include the figure from Essential Revisions #3 as a supplementary figure. Also include: a) the average firing rate traces for each of the 4 flashes, and b) confirm the behavioral effects for the low-frequency but not high-frequency stimulation.

*Reviewer #2:*

The authors have also performed the analysis relating to the NDMI and PDMI. They report a significant negative correlation, between the two. Looking at Figure 5 this seems to be driven by 5/42 experiments. They do not report what type of correlation was calculated (Pearson? Spearman? Robust correlation to control for outliers?). I think a robust correlation would be appropriate, while given the distribution of data I assume Pearson is inappropriate. They also present a line, which I assume is a slope of a linear regression? Given that NDMI and PDMI are dependent variables, slopes need to be calculated of x vs. y and y vs. x, and then the average slope needs to be taken.

---

## [Author Response]

Essential revisions:1) Anatomical evidence confirming that the neurons were actually transfected should be provided. This includes the information about the affected cell type, their layer locations and variability of the expression.

Unfortunately we cannot provide post mortem histology on the two animals used in this study. Monkey A is still being used for an ongoing experiment. Euthanizing Monkey A for histology would halt that experiment. Monkey C had to be sacrificed due to health complications and unfortunately the brain tissue at the recording site, which was penetrated multiple times during the experiment (and another subsequent experiment), was not in suitable condition for post mortem histology.

However, we have performed histological analysis in a third monkey, at a site not used for recording and this confirmed that expression of C1V1 and EYFP was biased almost exclusively to excitatory neurons at and around the injection site. See Author response image 1.

White arrows in the upper left panel indicate neurons that were immuno-positive for EYFP (FP+; green, top-left panel). White arrows in the upper right panel show neurons that were immune-positive for a cocktail of inhibitory neuron markers (parvalbumin, calretinin, or calbindin, red, top-right panel). The lower left panel shows a merged image illustrating that in this sample there was little or no overlap. Lower right panel shows quantification of this over 119 neurons imaged in five different fields of view. Only 1.7% of all EYFP-positive neurons counted of the 119 neurons were double labeled for both EYFP and PV/CR/CB (n = 2) (bottom-right panel), indicating that expression was heavily biased toward excitatory neurons. Scale bar, 20 mm. This is Figure 1B from Nassi et al., 2015, Neuron, which was conducted in our laboratory just prior to conducting this study using the identical viral construct used in the present study. We would be happy to include a version of this figure as a supplementary figure if the reviewers and Editor feel that this would strengthen the manuscript.

Reviewer 1: “Can you confirm that neurons were transfected?”

In the current study, we used the same viral constructs and the same viral injection protocol as was used in Nassi et al., where we quantified selective expression in pyramidal neurons. As in Nassi et al., 2015, in the present experiment we used fluorescent goggles to visualize protein expression at the injection site (See Figure 1C, main manuscript). We were able to see a bright fluorescent spot in both monkeys, throughout the experiment, indicating continued expression of GFP-tagged opsin. Additional evidence that cells were not only transfected but that they expressed opsin at a level sufficient to drive neurons is that we found modulation of firing that was tightly time locked to laser stimulation (see e.g., phase locking with laser intensity at high and low frequency: Figure 4—figure supplement 2) and firing rates that varied with levels of optogenetic stimulation (irradiance response curves, Figure 2—figure supplement 4). Taken together, all these lines of evidence give us a high level of confidence that not only were neurons transfected but also they continued to do so through the months of the study in both animals.

Reviewer 1: “Which types of neurons, and in which layers? How variable was the expression across layers?”

As we mentioned above, we find that, using this viral construct > 97% of transfected neurons are excitatory. There was no bias in expression toward any particular layer (Figure 1C of Nassi et al.).

Reviewer 1: “Can you rule out retrograde transfection from axon terminals?”

Lentivirus is primarily a locally infecting virus. Retrograde labeling with sufficient levels of opsin to be functional in vivo is very challenging and has been difficult to accomplish in the primate. So this possibility seems very unlikely.

Thus, in conclusion, based on our published data in V1 and similar in vivo epifluorescence profiles we observed in V4, we conclude that expression of opsin is primarily in excitatory neurons throughout all layers with a similar lateral spread (200300𝜇m). We would be happy to include supplemental figures from the data collected in the Nassi et al., 2015 study, showing this in the present paper if the reviewer and editors feel that this would strengthen the paper.

2) Variable performance on the orientation change task makes it difficult to interpret the effects of optogenetic stimulation. This problem should be addressed.

The monkeys used in our experiments were highly trained in the orientation change detection task and participated in multiple experiments. As a result, they could achieve performance levels that were very high even for the smallest orientation changes. The smallest orientation change that we tested was 1-degree orientation. This is a substantially lower threshold than has been achieved in primate studies (See e.g., Figure 1 from Cohen and Maunsell, 2009). However, we know from the psychophysical literature (Campbell and Green, 1965; Cowey and Rolls, 1974) that humans can detect orientation changes that are much lower than 1-degree, and that human observers are most sensitive to orientation changes for gratings presented near the cardinal orientations. We looked back at our data and found indeed a pattern consistent with this. For sessions in which the baseline (non-target) orientation was horizontal or vertical, monkeys consistently performed better in detecting orientation changes of 1 degree, the smallest change we tested. Please see Author response image 2, which shows% correct in detecting the target over eight sessions, including three (red stars) in which the non-target was presented at a cardinal orientation. This was a likely source of variation in animal performance in our study. Despite this source of variability, we find that low- but not high- frequency optogenetic stimulation systematically impaired performance for fine orientation discriminations at the stimulation site.

**Author response image 2. respfig2:** Examples of behavioral sessions without optogenetic stimulation. The red asterisks mark behavioral sessions where the baseline orientation (orientation of the non-target stimuli) was either horizontal or vertical.

3) The presence of on- and off- transients in visual stimulation occurring at a frequency of 3-5Hz creates a potential problem in the study aimed at detecting correlations occurring at similarly low frequency. It is important to rule out that optogenetic stimulation may be affecting the detection of the onsets and offsets of visual stimuli. Reviewer 1 suggests a control condition that would eliminate such transients to see whether optogenetic manipulation is still selective for low frequencies.A related question (reviewer 2) concerns the phase of optical stimulation relative to stimulus presentation whether the presence of the behavioral deficits depended on the phase of optical stimulation. Also, provide information about the phase of optical stimulation used to compute noise correlations

We agree that these are important potential confounds that need to be addressed. Though as noted above, it was, unfortunately, not possible to do the very thoughtfully considered control experiments in these animals, as reviewer 1 had suggested (one animal had to be humanely euthanized due to health concerns and the V4 chamber of the second animal has been removed and replaced with a V1 chamber). We have, however, undertaken analyses that we believe address these concerns. We reasoned that if the change in performance were due to a phase alignment between the visual stimuli and the optical stimulation, we would expect to find variation in behavioral performance that varied with laser phase (a point raised by reviewer 2). These analyses are described below, after each reviewer’s individual comments.

Reviewer 2: “it will be important to know whether behavioural deficits occurred equally across different phases, or whether they were co-modulated”

We have added a supplementary figure (Figure 4—figure supplement 3) in which we first show (panel A) that stimulus onset times were not phase locked to the laser, at either frequency. The distribution of target stimulus onset times is plotted as a function of laser phase for low-frequency (left panel) and high-frequency (right panel) optical simulation. This is due to the aperiodic nature of the stimulus flashes, with variable inter-stimulus-intervals between each non-target and between the final non-target and the target. So, targets were equally likely to occur at all phases of laser stimulation and there was therefore no tendency for targets to phase lock to the laser, at either frequency.

We next examined whether discrimination performance varied as a function of laser phase and target onset. See Panel B, which shows percent correct performance as a function of the phase of the laser at the time of target stimulus onset, grouped into 4 phase bins. We find no significant difference in performance as a function of laser phase either on low-frequency trials (Panel B) or high frequency trials (data now shown).

Reviewer 2: “The histograms show and the noise correlations also seem to be calculated over different optogenetic phases […] it will be important to determine whether there were changes in either when calculated for different optogenetic phases”.

As the reviewer fully appreciates, induction of correlation by optical stimulation is the method by which we introduced correlations in the experiment. However, if we are interpreting the reviewer’s concern correctly, s/he is making a more subtle point: that there might be phase-dependent variation in the degree of endogenously generated correlation. This would not be surprising, given that increases in luminance contrast have been found to reduce pairwise “noise” correlations. So, one might imagine that, as with contrast elevation, noise correlations might similarly fall off with laser intensity. We have therefore undertaken an analysis to test this directly, in which we compute pairwise correlations as a function of laser phase. This is shown in Author response image 3.

In this analysis, we examined the phase dependence of spike-count correlations for both low- and high-frequency correlations (200ms and 50ms counting windows, respectively) on low- and high-frequency optical stimulation trials. That is, for each pair of neurons, we computed the spike rate for each cycle of laser stimulation, divided into four phase bins. We then computed correlations across multiple cycles of laser stimulation. The results of this analysis appear in Author response image 3. Though it is not statistically significant, there does appear to be a tendency for spike count correlations to be lower, on average, at the peak of laser intensity. However, we find no evidence that perceptual performance varied with phase and so we do not believe this change in correlation, even if it were significant, influenced the monkey’s performance. Nor can it account for the observed perceptual impairment that was caused by low frequency laser stimulation, which was phase independent (Figure 4—figure supplement 3B).

**Author response image 3. respfig3:** Phase dependence of spike-count correlations for both low- (left panel, 200ms counting window) and high-frequency correlations (right pane, 50ms counting window) for the low-frequency optical stimulation condition.

4) Data presentation limited to averages and distributions does not allow the evaluation of significant effects in each experiment. The data suggest that in some cases optogenetic stimulation resulted in elevated thresholds and slopes and in some cases in opposite effects. This is a problem that needs to be addressed and discussed.

We do see evidence of both changes in threshold and changes in slope. To give the reviewer and reader a sense for the data, we have added a supplementary figure (Figure 2—figure supplement 5) in which we show the changes in behavioral performance for each experimental session. This figure shows that, consistent with our main conclusion, low frequency laser induced correlations did tend to increase threshold (preponderance of red and green points to the right of zero. There were also changes in slope, which tended to increase with laser stimulation (red and blue points above zero). In total, for a large fraction of the sessions, both the changes in threshold and slope were significant (20/42). 11/42 sessions had significant threshold change only, while 5/42 sessions had significant slope change only.

5) Please provide a direct comparison between firing rates with and without optical stimulation, showing effects on individual neurons. Rate modulation index does not allow the reader to assess such effects directly.

We have added a supplementary figure (Figure 4—figure supplement 1) in which we compare the spike rates for all neurons in our population between the optical stimulation conditions (both low- and high-frequency) and the baseline (unstimulated condition). We find no significant modulation of firing rate due to optical stimulation, as indicated by the points falling along the line of unity.

6) Was orientation tuning affected by optical stimulation? If they were, the disruption of orientation representation in V4 by stimulation could potentially explain the behavioral effects. This should be addressed.

The reviewer is correct to point out that if the laser altered tuning, this could affect the monkey’s performance and possibly account for the observed impairment. In particular, if the differences in firing rate between target and non-target orientations were reduced by low frequency optical stimulation, this could explain our findings. While we did not record tuning curves, we do have data from the non-target orientation and from multiple target orientations, so we could examine the effect of laser stimulation over the range of orientations spanned by targets and non-targets.

The results of this analysis are shown in a new supplementary figure (Figure 4—figure supplement 4), where we plot the response rates to the non-target and target orientations for the optical stimulation (for both low- and high-frequency) and unstimulated baseline conditions. To control for variation in mean firing rate across neurons, we have normalized the firing rates by the maximum rate for each neuron. We do not see any significant differences in firing rates to the different orientations due to optical stimulation.

7) Please explain cued and uncued locations vs. "catch trials" and their randomization during the experiment (reviewer 1).

We thank the reviewer for pointing out that we were not clear in our use of these terms. We have now revised the Materials and methods section to make this clear. To summarize: The time of target onset was drawn from an exponential distribution with time constant of 3 seconds, with the constraint that the target could not appear sooner than 1 sec after the start of the trial. If the target time drawn from the exponential distribution was >5 seconds, the trial was deemed a ‘catch’ trial and the target never appeared (13% of trials). On these catch trials, the monkey was rewarded for holding fixation to the 5 second mark. If the time drawn from the distribution fell between 1 second and 5 seconds, the target appeared at that time and the monkey was rewarded if it detected the target (“hit”) with an eye movement. Failure to detect the target was considered a “miss” and no reward was given. Trials were blocked according to which location was cued (left or right of fixation). When the target appeared, 95% of the time it appeared at the cued location. The remaining 5% of trials were deemed ‘foil’ trials.

The decision whether or not to stimulate (50/50), and the frequency of stimulation, were completely randomized from trial to trial.

Reviewer #1:[…] This is an ambitious set of experiments to attempt to pack into a short paper, especially given all of the potential pitfalls and controls that need to be considered. In the end, I was not fully convinced that the results as presented support the conclusions. There are several basic issues that need to be addressed to make the case more convincing. I also think that a more detailed unpacking of the data is need in order to understand the results.The most obvious missing piece is histology. Figure 1C shows a surface view of the cortex illustrating the expression of EYFP. This is not sufficient to answer the questions relevant for interpreting the behavioral and neuronal data. Can you confirm that neurons were transfected? Which types of neurons, and in which layers? How variable was the expression across layers? Can you rule out retrograde transfection from axon terminals? Without histology to verify what was stimulated, I find it difficult to interpret the results.

Please see our responses to Essential revisions #1 above.

Performance in the orientation change task seems extremely variable, to the point that it raises concerns about how to interpret changes in task performance. In some cases, the thresholds for detecting orientation changes seem to be in the expected range (a few degrees) – for example, Figure 2B and Figure 2—figure supplement 2A. But in other cases, the thresholds are unusually high, 10 degrees or more (Figure 2—figure supplement 2C). There are also unexpected elevations in% correct for signal values that should be below threshold – for example, 40% correct for the lowest orientation change value in Figure 2—figure supplement 2B. How could the monkey get 40% correct for an undetectable signal? In order to interpret the optogenetic manipulation, I would need to be reassured about the reliability of task performance in the absence of optogenetic manipulation.

Please see our responses to Essential revisions #2 above.

There is a very basic aspect of the experimental design that seems like a problem, but perhaps the authors have a very well-reasoned explanation for this approach. The hypothesis is that low-frequency (4-5 Hz) correlations play a central role in cortical processing, and this guides the choice of sinusoidal optogenetic stimulation. But the stimulus itself is flashed on for 200 ms and then left off for 200-400 ms, which means that there were visual transients (on or off) also occurring at a frequency of 3-5 Hz. If the goal is to test the importance of intrinsic low-frequency correlations, why use a stimulus that includes transients in this same frequency range? It seems that an alternative explanation for the perturbations in performance might be that the optogenetic stimulation masks the ability to detect the onsets and offsets of the visual stimulus. Can this be ruled out? If the stimulus were simply left on for an extended period of time (seconds), and then changed orientation, would the optogenetic manipulation still be effective and still selective for low frequencies?

Please see our responses to Essential revisions #3 above.

The data were compressed for presentation in ways that made it difficult to understand what additional significant effects might be present in each experiment. Figure 2C is a good example of this. The histogram illustrates that, overall, both the threshold and slope increased in trials with low-frequency optical stimulation, by showing that the population-level ratio of stimulation relative to baseline is significantly greater than 1. First, please clarify the stats (here and elsewhere). This is reported as a t-test in the fourth paragraph of the Results and Discussion. Which type – paired sample? Is the distribution normal, or should a non-parametric test be used? Second, please report the data in a way that allows us to appreciate what happened on individual experiments. Given the scale on the x-axis, it looks like optical stimulation may have caused significant increases in threshold or slope in some experiments and significant decreases in others. If so, it would be misleading to base the conclusion on the average effect. This point is especially relevant for the attend away condition (Figure 2D), in which the average threshold is not different, but the spread in the histogram suggests that many individual experiments were significant but roughly equally split between increases and decreases.

Please see our response to Essential revisions #4 above.

I had similar concerns about the neuronal data in Figure 4. In addition to plotting the rate modulation index, please directly compare the firing rates with and without optical stimulation. Was the firing rate significantly changed for individual neurons? The effects of attend-in and attend-away should be similarly documented across the population of neurons. Did optical stimulation significantly change this modulation for individual neurons? The concern here is that there may be no difference on average, but a more or less balanced combination of significant increases and significant decreases.

Please see our response to Essential revisions #5 above.

Beyond changes in firing rate, it is also possible that the optical stimulation disrupted the tuning properties of the neurons. Do you have data to confirm that the orientation tuning and receptive field properties of the neurons was not changed during optical stimulation? If the claim is specifically about the role of low-frequency correlations, it is important to rule out the possibility that behavioral effects were due simply to disrupting the representation of orientation information in V4.

Please see our response to Essential revisions #6 above.

Reviewer #2:[…] I was unable to determine whether the phase of stimulation was fixed relative to stimulus presentation. Probably not as stimulus presentation varied within a trial. This means that orientation changes would occur at variable phases of optogenetic stimulation. If so, it will be important to know whether behavioural deficits occurred equally across different phases, or whether they were co-modulated.

Please see our responses to Essential revisions #3 above.

The histograms show and the noise correlations also seem to be calculated over different optogenetic phases (if my assumption from above is correct). It will be important to determine whether there were changes in either when calculated for different optogenetic phases. If so, these need to be documented in detail.

Please see our responses to Essential revisions #3 above.

The authors state that no effect was seen in the attend away condition. Was that even true on catch trials (i.e. attend away, but change occurs at RF)? If so, it needs to be explained why.

We think that the reviewer is referring to what we termed “foil” trials in our manuscript (i.e. change occurs in uncued spatial location). Since the foil trials were roughly 5% of the trials, which were then split between the stimulated and unstimulated conditions, we did not have sufficient statistical power to evaluate the effect of optical stimulation during the foil trials in the attend away condition.

Why was the higher frequency in the beta range, not in the gamma range, where some people might expect to see behavioural benefits to occur?

One of our initial hypotheses was that gamma range stimulation might produce behavioral benefits. However, in our pilot experiments, we tried gamma range stimulation using both sinusoidal stimulation and ramp stimulation (Adesnik and Scanziani, 2010, Nature) protocols, but we did not find any noticeable change in behavior. For our main experiment, we used the beta frequency range for our control condition, since it was above the 10Hz frequency range where a previous study from our lab (Mitchell, Sundberg and Reynolds, 2009) did not find an appreciable change in attention dependent coherent activity.

[Editors' note: further revisions were requested prior to acceptance, as described below.]1) Lack of histologyThe reviewers felt that the lack of histology seriously complicates the interpretation of the results but are willing to accept a possibility that the expression patterns of the viral vector in V4 are the same as those shown for V1 in the previous publication from the lab. However, you should add appropriate disclaimers about the assumptions about transfected neurons (based on the published V1 data), and discuss how your interpretation would be affected if these assumptions turn out to be incorrect.

We acknowledge that it would have been more compelling to perform histology in order to demonstrate the viral expression pattern. As we have indicated in the previous round of review, one of our animals who had to be euthanized due to health complications but we could not perform histology since the dura mater adhered to V4 bilaterally. The second animal is still in use for experiments. However, we feel reasonably confident that the expression pattern in V4 would be similar to what has been seen in V1 (Nassi et al., 2015) and perirhinal cortex (Tamura et al., 2017), for example.

We have added a paragraph (Results and Discussion, ninth paragraph) in the revised manuscript outlining these details.

2) Variability in orientation thresholdsThe reviewers felt that the explanation attributing variability to the difference in thresholds for oblique and cardinal orientation, needs stronger documentation. To that end, rather than providing 8 example psychometric functions, the relationship between base orientation and performance should be documented by plotting thresholds for each of the sessions as a function of orientation.

We have added new analysis which shows that monkeys performed the task better during sessions in which they were required to discriminate orientation changes from cardinal (0°, 90°) non-target orientations, as reflected in elevated performance in detecting the smallest orientation change and elevated threshold for cardinal compared to non-cardinal orientations (Figure 2—figure supplement 2D).

We have added additional text to the second paragraph of the subsection “Data analysis”.

3) Potential confound of visual and optogenetic stimulation delivered at similar low frequenciesReviewers did not feel that the phase analysis included in the revision adequately addressed the problem. Reviewer 2 commented that while the overall effect may not be phase-specific, it may be specific "to cases where the optogenetic frequency is similar to the stimulus onset/offset frequency". Please provide additional information, to address this.

We apologize for misunderstanding the reviewer’s initial concern, which we took to be centered on phase. It does seem plausible that neural and perceptual sensitivity might vary with the depolarization state of the neurons, and might therefore vary with laser phase. From the reviewer’s comments, he or she appears to agree that we do show that the target stimuli are not locked to any particular phase of the laser (Figure 4—figure supplement 3A) and that behavioral performance does not depend on when the target stimulus appeared in the laser cycle (Figure 4—figure supplement 3B). We have carefully considered the reviewer’s suggestion that the effect of low frequency stimulation might selectively impair stimuli presented at a similar frequency. Although we are unaware of any studies that have examined this, one might imagine that presenting the laser at a particular frequency might cause a sort of frequency-dependent adaptation that would cause neurons to be less sensitive to visual stimuli presented at a similar frequency. If so, the low (4-5Hz) frequency laser could reduce the responses evoked by 200 msec stimuli, impairing the monkey’s ability to discriminate the stimuli, while the high frequency laser might not cause this effect, explaining the observed impairment. To test this, we measured the firing rates evoked by the first four non-target stimulus flashes, on no-laser, low-frequency-laser and high-frequency laser trials, all on trials when the monkeys were attending in to the stimuli appearing at the location corresponding to the opsin site. Though the first stimulus in the sequence evoked a stronger response than the subsequent stimuli (reflecting a form of visual-stimulus-driven adaptation), we find no evidence that the addition of the laser at either frequency caused a change in mean firing rate. Please see Author response image 4 (n=38 units; mean +/- s.e.m.). We hope we have correctly understood the nature of the reviewer’s concern. If the reviewer and the editors feel that this analysis will strengthen the paper, we will be happy to include this as a supplementary figure.

**Author response image 4. respfig4:** 

Please address the absence of label for the x-axis in Figure 4—figure supplement 3B and different sign of effects on threshold and sensitivity. One of the reviewers suggests sorting the experiments into subsets based on how the psychometric curves change (for example, based on where the experiments fall in Figure 2—figure supplement 5).

We fixed the label in Figure 4—figure supplement 3B.

The second part of the comment is addressed as part of #4 below.

4) Changes in thresholds and slopesThe summary plot (Figure 2—figure supplement 5) reveals not only increases in threshold (22/42 cases) but also decreases in threshold (~14 cases). Please address the apparently opposite effects produced by stimulation.In addition, it appears that stimulation increased the slope more often than decrease it. Since the increase in slope is likely to be associated with an increase in sensitivity, the proposed role of decreased correlations in improving the separability of neuronal activity is puzzling. This effect appears inconsistent with the proposed interpretation of the changes in spike-count correlations with stimulation. The reviewers suggest that you link the measured changes in correlations with the observed changes in psychometric functions by using a decoding scheme similar to that used by Cohen and Maunsell, 2009. This approach would allow you to predict the changes in psychometric curves given the changes in spike-count correlations.

We thank the reviewers for this excellent suggestion. We have now done this analysis and find that for recording sessions where perceptual discrimination was impaired by low frequency laser stimulation, the laser reduced discriminability of target and nontarget at the neural level, as well. In this analysis, which appears in new Figure 5, we calculated a neural measure of discriminability (neural discriminability modulation index, NDMI) that was modeled on the analysis used by Cohen and Maunsell, 2009. We computed this measure in two ways: (1) using the full N-dimensional space that was defined by the responses across the N neurons recorded on each recording session and (2) a second pairwise analysis. Each of these analyses provided a measure of the degree of overlap between the neural responses evoked by the discriminanda. We repeated this analysis for no-laser trials and laser trials, and computed an index whose value corresponded to the change in discriminability caused by the addition of the laser. For each session, we then compared the change in perceptual threshold caused by the laser (we have termed this change the perceptual discrimination modulation index, PDMI, in the revised manuscript), with these measures of laser-induced changes in neural discriminability. This analysis revealed that on sessions for which the laser impaired perceptual discrimination at the behavioral level, there was a marked reduction in discriminability at the neural level, for both neural measures. The NDMI and PDMI are also significantly negatively correlated, using either NDMI measure. In other words, laser-induced reductions in discriminability at the neural level corresponded to increased perceptual thresholds.

We also repeated the analysis for laser induced changes in slope. For the purpose of this experiment, the psychometric function threshold is the measure of interest. It is a measure of the change in orientation needed to reliably discriminate target from nontarget. Slope does not measure this, and though we do see laser dependent changes in slope, they were not correlated with the NDMI. A change in slope means that the shift from detecting to not-detecting occurs over a narrower range of orientation changes, which is not really the point of the paper. We also note that the slope is very sensitive to the form of the psychometric function that is fit to the data. Instead of fitting a logistic function to the data as we have, if we fit a Weibull function to the data, we see a

similarly significant increase in threshold, but no significant change in slope.

Thus, we have focused on the threshold change in the manuscript and report the slope change in a supplementary figure (Figure 2—figure supplement 5A).

The NDMI analysis is detailed in the sixth paragraph of the Results and Discussion and in the subsection “Neural Discrimination Modulation Index (NDMI)”.

5) Orientation tuningPlease provide additional information concerning the Figure 4—figure supplement 4 scaling on the x-axis and pooling data across neurons. How were the neurons used for this analysis selected? (see comment from reviewer 1).

We realized that our analysis in the previous round was incorrect in the sense that the tuning curves across neurons were not aligned to peak response. We have corrected this (see updated Figure 4—figure supplement 4). The spike rates were normalized to peak response across stimulation conditions (optical stimulation and baseline no-stimulation conditions). The population tuning curves were plotted after aligning the peak response of each neuron to zero and binning the data into orientation bins.

6) Please provide documentation of false alarm rates from catch trials with and without optogenetic stimulation (see comment from reviewer 1).

We now report false alarm rates for both optical stimulation conditions and compare them to the false alarm rates to the baseline condition (Figure 2—figure supplement 2E). We have added text to document this: “Nor did we find significant changes in false-alarm rates with either low- or high-frequency stimulation (𝑝 > 0.1, *t*-test; Figure 2—figure supplement 2E). This was true for false alarms made during catch trials as well as on non-catch trials. Thus, we find no evidence that laser stimulation caused our subjects to mis-perceive a non-target as a target.”

As mentioned, false alarm rates on catch trials are similar to those on non-catch trials: ~6% of trials. The ~50% performance rate in catch trials reported in the previous version of the figure (Figure 2—figure supplement 2A, ‘*’ symbol) arises mostly due to the monkey breaking fixation toward the end of the long catch trials. We realize that this way of reporting performance in catch trials could be a potential source of confusion. We have rectified this by calculating catch trial performance after first excluding errors due to fixation breaks and have updated the figure.

7) The phase locking of spikes to the phase of stimulation should also be shown during the stimulus, in addition in the absence of stimulation.

We have done this additional analysis and included the result in Figure 4F, with additional text: “We see a similar phase locking due to optical stimulation during the stimulus presentation period (Figure 4F, 𝑝 < 0.01, Rayleigh test), although the peak of the phase-lock distribution for the stimulus presentation period occurs earlier (around 120°) compared to that for the pre-stimulus period (around 210°). This would be expected, as the neurons are depolarized by the visual stimulus and hence more easily pushed to spiking threshold by optogenetic depolarization, as compared to when no stimulus is present.”

8) The effect of optogenetic phase on behavior should be aligned with the preferential population activity, which seems to peak/trough at 240° and 60° respectively.

We have added another sub-panel in Figure 4—figure supplement 3B where we have done the phase contingent analysis using a different bin arrangement as per the review’s suggestion.

9) Please clarify what statistical test was used to determine that the phase of optical stimulation did not affect behavior or orientation tuning

We have added the details of the statistical test in the fifth paragraph of the Results and Discussion.

Reviewer #1:The authors do not have histology for the two monkeys used in the study but they show a figure from a previously published study using the same viral vector injected into cortical area V1.Will the expression patterns be the same in V4, the area targeted in this study? I don't know, and I don't know of any published work using this vector in macaque V4. Apparently, there is tissue from one of the monkeys (that was sacrificed) from this study; even if the tissue is damaged due to penetrations it should be possible to identify transfected cells, and the approximate size and layer distribution. It is not clear what steps were taken along these lines, or if that tissue was unfortunately discarded.Please see our response to Essential revisions #1.The authors explain that this is probably due to differences in detection performance – in particular, thresholds were lower for baseline orientations near the cardinal (i.e., horizontal and vertical) orientations. They show 8 sample psychometric curves that are consistent with this explanation.This seems plausible, but the author should show it holds true for the other 34 sessions as well. If you simply plot threshold for each session as a function of the baseline orientation, this would show whether the variance was indeed systematically related to the baseline orientation.Please see our response to Essential revisions #2.The authors agree these are important potential confounds but for technical reasons, they are not able to do the control experiment in which the stimulus is simply left on, to test whether the optogenetic stimulus might be masking the visual onset and offsets that occur in the same frequency range.The phase analysis is interesting, but does not address the same point exactly. I wouldn't necessarily expect the effect to be phase-specific, but I do suspect it might be specific to cases where the opto frequency is similar to the stimulus onset/offset frequency. This question remains open.

Please see our response to Essential revisions #3.

Some comments about Figure Figure 3—figure supplement 3B: What is the x-axis and why is it unlabeled? There seem to be some interesting possible mixed effects at low delta orientations. If you pool across all sessions (with different sign of effects on threshold and sensitivity) perhaps some effects are getting averaged out. Have you tried sorting the experiments into subsets based on how the psychometric curves change (for example, based on where the experiments fall in Figure 2—figure supplement 5)?Please see our responses to Essential revisions #3 and #4.The authors now provide a summary plot (Figure 2—figure supplement 5) that summarizes the changes in threshold and slope. This is helpful. It shows that in addition to the main effect reported in the paper – the increase in threshold seen in 22/42 cases – there is also sometimes a significant decrease in threshold (~14 cases). Any ideas about why the effect flips sign in these cases?More curiously, and harder to understand, the stimulation also tends to increase the slopes more often than it significantly decreases the slopes. An increase in slope would imply that the sensitivity of the monkey during the stimulation had increased. Given the proposed role of decreased correlations in improving the separability of neuronal activity, shouldn't the main effect have been a decrease in slope?Please see our response to Essential revisions #4.The authors respond that they did not measure tuning curves. However, they do have data from the non-target orientation and some target orientations, which they report in Figure 4—figure supplement 4. I find it difficult to evaluate this plot because I don't understand the scaling on the x-axis or how data were pooled across neurons. It is also not clear that data from all neurons should be included in this analysis, unless their activity was strongly modulated across the range of orientations used (i.e., the data indicate direction tuning over the domain tested). And then the data might be aligned on the x-axis so that 1 value corresponded to the "best" direction.The issue of possible changes in neuronal tuning is critical for interpreting the results. Perhaps the authors can do more to address this.Please see our response to Essential revisions #5.The authors now explain their definition of 'catch' trials.Are the FA rates on from catch trials documented somewhere in the paper? I did not see it except for Figure 2—figure supplement 2A, which curiously appears to show a false alarm rate on catch trials of about 50% Is this correct? How can the FA rate be that high when the hit rate drops well below that for small orientation changes? I would expect the FA rate to be the floor for the curve.Aside from trying to understand the plots, the other reason for asking about FAs is to know whether the FAs also changed with optogenetic stimulation. This would be important for assessing possible changes in response criterion, which would also be important to nail down, since changes in criterion could also shift the psychometric curves.

Please see our response to Essential revisions #6.

Reviewer #2:They describe the phase locking of spikes to the phase of stimulation when no stimulus was present. However, it would be important to see this also for the stimulus period.

Please see our response to Essential revisions #7.

The authors use 4 bins to calculate the effect of optogenetic phase on behaviour, but these are not aligned with the preferential population activity alignment, which seems to peak/trough at 240° and 60° respectively. This needs to be done.

Please see our response to Essential revisions #8.

It is unclear what statistical test was used to determine whether behaviour was unaffected by the phase of optical stimulation?

Please see our response to Essential revisions #9.

The same is true for the effect on orientation tuning.

Please see our response to Essential revisions #9.

In general, statistical reporting should be checked and adequately improved.[Editors' note: further revisions were requested prior to acceptance, as described below.]1) Please substitute multiple t-tests with ANOVA to reveal potential interactions.

We have included ANOVA analyses to show:

a) Behavioral performance was not dependent on laser phase (no main effect of laser phase, no significant interaction between laser phase and delta orientation (the trial-by-trial difference between target and non-target orientation); Results and Discussion, fifth paragraph).

b) Orientation tuning curves were not altered by laser stimulation (no main effect of laser condition, no significant interaction between laser condition and orientation; Results and Discussion, fifth paragraph).

c) Visual stimulus driven adaptation is not modulated by laser simulation (no main effect of laser condition, no significant interaction between laser condition and stimulus position within trial; Results and Discussion, fifth paragraph).

2) Please address the point concerning correction analysis, raised by reviewer 2.

As suggested by the reviewer, we have repeated the analysis using robust regression which excludes outliers in the data. In our dataset, the approach did not find any outliers, so the results were unchanged (see Author response image 5; red and green lines completely overlap). The reviewer has also rightly pointed out that since both NDMI and PDMI are dependent measures, a line fitted from a Model II regression is the appropriate approach. We have updated the fitted lines in Figure 5B with those obtained from a Model II regression (also shown in Author response image 5; black line).

**Author response image 5. respfig5:** 

3) Please include the figure from Essential Revisions #3 as a supplementary figure. Also include: a) the average firing rate traces for each of the 4 flashes, and b) confirm the behavioral effects for the low-frequency but not high-frequency stimulation.

We have included a new supplementary figure (Figure 4—figure supplement 5) as suggested by the reviewer. The results show that optogenetic stimulation at either frequency altered neither mean firing rates (as we intended) nor had a measurable effect on adaptation. We have included the firing rate traces for each flash. We have commented on the behavioral effects in the main text (Results and Discussion, fifth paragraph).

Reviewer #2:The authors have also performed the analysis relating to the NDMI and PDMI. They report a significant negative correlation, between the two. Looking at Figure 5 this seems to be driven by 5/42 experiments. They do not report what type of correlation was calculated (Pearson? Spearman? Robust correlation to control for outliers?). I think a robust correlation would be appropriate, while given the distribution of data I assume Pearson is inappropriate. They also present a line, which I assume is a slope of a linear regression? Given that NDMI and PDMI are dependent variables, slopes need to be calculated of x vs. y and y vs. x, and then the average slope needs to be taken.

We thank the reviewer for raising these issues. We repeated the analysis using robust regression, which excludes outliers. Robust regression did not identify and exclude any data points, so the results are unchanged. We also repeated the regression analysis using Model II regression, which treats both variables as dependent variables. Please see our response to Essential revision #2 above.